# A General-Purpose Theorem for High-Probability Bounds of Stochastic Approximation with Polyak Averaging

**Sajad Khodadadian**[*]
Grado Department of Industrial and Systems Engineering
Virginia Polytechnic Institute and State University
Blacksburg, VA 24061
`sajadk@vt.edu`

**Martin Zubeldia**[*]
Department of Industrial and Systems Engineering
University of Minnesota
Minneapolis, MN 55455
`zubeldia@umn.edu`

## Abstract

Polyak–Ruppert averaging is a widely used technique to achieve the optimal asymptotic variance of stochastic approximation (SA) algorithms, yet its high-probability performance guarantees remain underexplored in general settings. In this paper, we present a general framework for establishing non-asymptotic concentration bounds for the error of averaged SA iterates. Our approach assumes access to individual concentration bounds for the unaveraged iterates and yields a sharp bound on the averaged iterates. We also construct an example, showing the tightness of our result up to constant multiplicative factors. As direct applications, we derive tight concentration bounds for contractive SA algorithms and for algorithms such as temporal difference learning and $Q$-learning with averaging, obtaining new bounds in settings where traditional analysis is challenging.

## 1 Introduction

Stochastic approximation (SA) algorithms are central to modern machine learning and optimization, serving as the foundation for methods ranging from stochastic gradient descent to Reinforcement Learning (RL) algorithms such as temporal difference (TD) learning and $Q$-learning. The general form of an SA algorithm is given by

$$x_{k+1} = (1 - \alpha_k)x_k + \alpha_k F(x_k, w_{k+1}), \tag{1.1}$$

where $\alpha_k = \alpha/(k + h)^\xi$ for some $h > 1$, $\alpha > 0$, and $\xi \in [0, 1]$ is the step size, and $w_{k+1}$ represents random noise, possibly arising from sampling or environmental interactions. A significant body of work has been devoted to analyzing the convergence of such algorithms, providing both mean-square and high-probability error bounds under various assumptions (e.g., [49, 12, 19]).

---

[*]Equal contribution.

39th Conference on Neural Information Processing Systems (NeurIPS 2025).

A key technique for improving the asymptotic variance and robustness of SA algorithms is *Polyak-Ruppert averaging*, where instead of using the iterates $x_k$ directly, one uses the averaged sequence

$$y_k = \frac{1}{k+1} \sum_{i=0}^{k} x_i. \tag{1.2}$$

This averaging has been shown to achieve optimal asymptotic behavior in a variety of settings, including stochastic gradient descent [40, 37, 1], contractive SA [10], and RL algorithms [36, 33]. Despite its empirical and theoretical effectiveness, sharp finite-time high-probability bounds for averaged SA in general settings remain underexplored.

## 1.1 Our Contributions

In this paper, we develop a general-purpose framework for deriving *finite-time high-probability bounds* for averaged SA iterates. Specifically, we assume that the base SA iterates $x_i$, with probability at least $1 - \delta'$, satisfies a high-probability bound of the form $\|x_i - x^*\|_c^2 \leq \alpha_i f(\delta', k)$ for all $\delta' \in (0, 1)$ and $0 \leq i \leq k$, where $x^*$ is the limit point (e.g., the fixed point or optimum), and $f$ is a known function of $\delta'$ and $k$. Building on this assumption, we establish that for any $\delta \in (0, 1)$, with probability at least $1 - \delta$, the averaged iterates satisfy the bound

$$\|y_k - x^*\|_c^2 \leq c \frac{\log(1/\delta) + d}{k+1} \bar{\sigma}^2 + o\left(\frac{\log(1/\delta)}{k}\right),$$

where $d$ is the dimension of the variables, $\bar{\sigma}^2$ is a variance proxy, and $c$ is a universal constant. We also construct an example to show that our result is tight up to the constant factor $c$.

Our general theorem has several important consequences. First, we provide a principled way to select the step size schedule $\{\alpha_k\}_{k \geq 0}$ to optimize the concentration of averaged SA under a variety of operators $F$. Second, we apply our framework to *contractive SA algorithms*, establishing sharp concentration bounds in these settings. Third, we leverage our results to derive new high-probability bounds on the convergence of averaged TD-learning improving upon the prior result. Furthermore, we establish concentration bounds on the convergence of averaged $Q$-learning and off-policy TD-learning, which to the best of our knowledge are the first in the literature. Finally, the general nature of our framework allows it to be used to obtain high-probability bounds of averaged SA iterates based on future refinements of non-average SA bounds (e.g., if high-probability bounds for SA under more general assumptions are developed in the future).

## 2 Related Work

**Finite-Sample Guarantees for Contractive SA.** Recent advances have significantly improved finite-time analyses of SA algorithms in contractive and strongly-monotone settings, often driven by applications in RL and optimization. Early finite-sample mean-square error bounds were established for specific SA instances such as TD-learning [6, 15, 16, 26, 34, 32, 17, 35] and $Q$-learning [20, 3], but lacked tail probability guarantees. Subsequent work provided high-probability concentration results. For instance, [49] introduced non-asymptotic bounds for unprojected linear SA, and [41] analyzed asynchronous $Q$-learning. Using martingale concentration techniques [21, 23], [51] obtained an $\mathcal{O}(k^{-1/2})$ tail bound, while [12] utilized Lyapunov methods with convex envelopes to achieve finite-time results. This research culminated in general frameworks for Markovian SA: [13] provides polynomial tail bounds applicable to nonlinear contractive SA with Markovian noise. In particular, [14] (also [9, 8]) showed subgaussian concentration under minimal noise assumptions via exponential supermartingales, extending results beyond the i.i.d. setting. Sharper concentration bounds emerged for linear SA (LSA). [19] derived tight $\mathcal{O}(1/k)$ error rates for constant step size LSA. In nonlinear SA, notably $Q$-learning, averaging enhances performance. [53] established exponential tail bounds for averaged $Q$-learning, while [28] (also [30]) derived functional CLTs and optimal non-asymptotic error bounds for synchronous averaged $Q$-learning. Similarly, [55] demonstrated minimax-optimal complexity using averaging in variance-reduced $Q$-learning. Addressing practical concerns, [45] showed universal step sizes combined with tail-averaging yield near-optimal guarantees for linear TD-learning without projections or feature knowledge.

**Polyak-Ruppert Averaging in Stochastic Approximation.** Averaging is a classical variance-reduction technique in SA, known to improve the asymptotic efficiency of iterates [40]. For TD-learning with linear function approximation, [31] provide finite-time high-probability bounds for averaging, albeit their step size depends on the confidence bound. [39, 46] strengthen this by showing that tail-averaged TD-learning with constant step sizes achieves optimal-order bias and variance, without requiring step size tuning. Furthermore, [36, 18] establish similar results for linear SA, which generalizes TD-learning with linear function approximation. In addition, via simulation, [24] provides a negative result on the convergence of averaged TD-learning. In $Q$-learning and other nonlinear SA settings, [29] establish tight sample complexity bounds. Follow-up work shows Polyak-averaged $Q$-learning achieves asymptotic efficiency [33]. [33] also provides a statistical analysis showing convergence to the minimax lower bound for tabular $Q$-learning. [48] derive a non-asymptotic CLT using Stein's method, applied to TD-learning. [44] prove Berry–Esseen bounds and establish bootstrap validity for linear SA. These enable rigorous confidence intervals in reinforcement learning. In actor-critic and other two-time-scale SA, [25] prove the first non-asymptotic CLT under averaging. [7] and [10] obtain finite-time bounds, showing that averaging improves rates from $\mathcal{O}(k^{-2/3})$ to $\mathcal{O}(k^{-1})$.

**High probability bounds vs bound on the distribution:** Most of the prior work on the concentration of SA considers a step size which depends on the target probability $\delta$ [33, 46, 29]. Such analysis only provides a bound on a single point in the distribution of error. In contrast, in this paper, our step size does not depend on $\delta$, and we establish a bound on the entire distribution of the error.

## 3   Stochastic Approximation and Polyak-Ruppert Averaging

In this section we review the classical SA framework, and the averaging technique.

**Classical Stochastic Approximation**: Let $\bar{F} : \mathbb{R}^d \to \mathbb{R}^d$ be a (deterministic) operator, which has at least one fixed point $x^*$ (i.e., $\bar{F}(x^*) = x^*$). The goal is to find this fixed point. If we have access to the deterministic operator $\bar{F}$, under certain conditions (for instance, contractiveness of $\bar{F}$ or $\bar{F}(x) = x - \nabla h(x)$ for some non-convex function $h$) the iteration $x_{k+1} = \bar{F}(x_k)$ converges to a fixed point. However, in many applications we do not have access to the deterministic operator $\bar{F}$, and we only have access to noisy evaluations $F(\cdot, w_{k+1})$ of this operator such that $\bar{F}(\cdot) = \mathbb{E}[F(\cdot, w_{k+1}) \mid \mathcal{F}_k]$, where $\mathcal{F}_k$ is the sigma algebra generated by $\{w_1, w_2, \ldots, w_k\}$. The Robbins–Monro recursion [42] defines a sequence $\{x_k\}_{k \geq 0}$ as in Equation (1.1), where $\{\alpha_k\}_{k \geq 0}$ is a step size sequence. Under some regularity conditions on the operator and the noise, and with the step size conditions $\sum_{k=0}^{\infty} \alpha_k = \infty$ and $\sum_{k=0}^{\infty} \alpha_k^2 < \infty$, it is known that the iterates converge to $x^*$ almost surely [4]. SA recursions of the form of Equation (1.1) underpin a large class of algorithms in RL [50, 54] and optimization [22].

**Polyak-Ruppert Averaging**: This method was proposed independently by Ruppert and by Polyak and Juditsky [43, 40] to sharpen the statistical performance of SA. Given the raw iterates $\{x_k\}_{k \geq 0}$ defined by Equation (1.1), Polyak-Ruppert averaging defines the averaged iterates $\{y_k\}_{k \geq 0}$ as in Equation (1.2). It can be shown that $\sqrt{k}(y_k - x^*)$ converges asymptotically to a normal distribution which has a covariance matrix that matches the Cramér–Rao lower bound [40]. An advantage of this averaging technique is that we can achieve the best asymptotic covariance with a robust choice of step size $\alpha_k$ [38]. Besides the asymptotic results, there has been a long literature on the finite time analysis of SA algorithms, some of which are described in Section 2. Generally speaking, finite time bounds can be categorized into moment bounds (i.e., for some $m \geq 1$, finding a bound on $\mathbb{E}[\|x_k - x^*\|^m]$ or $\mathbb{E}[\|y_k - x^*\|^m]$ as a function of $k$) and high-probability bounds (i.e., finding a bound on $\mathbb{P}(\|x_k - x^*\| \geq \epsilon)$ or $\mathbb{P}(\|y_k - x^*\| \geq \epsilon)$ as a function of $\epsilon > 0$ and $k$).

## 4   Main Result

In this section, we present our main result. Throughout this section, we fix a given norm $\|\cdot\|_c$ which satisfies the following assumption.

**Assumption 4.1.** The function $\|\cdot\|_c^2$ is $M$-smooth with respect to the $\|\cdot\|_c$ norm, i.e., for all $a, b \in \mathbb{R}^d$, we have

$$\|a + b\|_c^2 \leq \|a\|_c^2 + \langle \nabla \|a\|_c^2, b \rangle + \frac{M}{2}\|b\|_c^2.$$

*Remark.* If the function $\| \cdot \|_c^2$ is non-smooth (e.g., if $\| \cdot \|_c = \| \cdot \|_\infty$), we can employ the machinery of the Moreau envelope [12] to obtain arbitrarily close smooth approximations of $\| \cdot \|_c^2$. Therefore, Assumption 4.1 is without loss of generality.

Next, we impose an assumption on the operator $\bar{F}$ and its noisy version $F$, and on their Jacobians $J_{\bar{F}}(x) := \frac{\partial \bar{F}(x)}{\partial x}$ and $J_{F_w}(x, w) := \frac{\partial F(x,w)}{\partial x}$.

**Assumption 4.2.** We assume the following.

  **(i):** The operator $\bar{F}$ admits at least one fixed point $x^*$, i.e., $\bar{F}(x^*) = x^*$

  **(ii):** The operator $\bar{F}$ is differentiable, and the matrix $J_{\bar{F}}(x^*) - I$ is invertible. Hence, we have

$$\nu := \min_{z \in \mathbb{R}^d} \frac{\|(J_{\bar{F}}(x^*) - I)z\|_c}{\|z\|_c} > 0.$$

  **(iii):** The operator $F(x, w) - x$ is $(N, R)$-locally psuedo smooth with respect to $\| \cdot \|_c$-norm. That is, there exists a radius $R > 0$ such that, for all $x$ satisfying $\|x - x^*\|_c \leq R$, we have:

$$\|(J_{F_w}(x^*, w) - I)(x - x^*) - (F(x, w) - x) + (F(x^*, w) - x^*)\|_c \leq N\|x^* - x\|_c^2.$$

Note that Assumption 4.2(i) is relatively weak and it is satisfied for a wide range of operators. For instance, this assumption is satisfied for any contractive operator. Moreover, Lemma C.3 implies that Assumption 4.2(ii) is also satisfied for contractive operators. Furthermore, it can be readily verified that the linear operator $\bar{F}(x) = Ax + b$ satisfies Assumption 4.2(ii), provided that $A$ is Hurwitz. Also, note that Assumption 4.2(iii) generalizes the notion of smoothness [2]. In fact, any smooth operator satisfies this assumption with $R = \infty$.

Next, we impose an assumption on the noise of the operator at any fixed point $x^*$.

**Assumption 4.3.** For any $d$-dimensional, $\mathcal{F}_k$-measurable random vector $v$, and for any fixed point $x^*$, we have

$$\mathbb{E}\left[\exp\left(\lambda\langle F(x^*, w_{k+1}) - \bar{F}(x^*), v\rangle\right)\big|\mathcal{F}_k\right] \leq \exp\left(\frac{\lambda^2 \bar{\sigma}^2 \|v\|_c^2}{2}\right), \qquad \forall \lambda \geq 0. \qquad (4.1)$$

Moreover, for any $d$-dimensional, $\mathcal{F}_k$-measurable random vectors $a$ and $b$, we have

$$\mathbb{E}[\exp \lambda\langle(J_{F_w}(x^*, w_{k+1}) - J_{\bar{F}}(x^*))a, b\rangle|\mathcal{F}_k] \leq \exp\left(\frac{\lambda^2 \hat{\sigma}^2 \|a\|_c^2 \|b\|_c^2}{2}\right), \qquad \forall \lambda \geq 0. \qquad (4.2)$$

*Remark.* Assumption 4.3 is equivalent to assuming the noise in the operator and the Jacobian are subgaussian [53]. An example that satisfies Assumption 4.3 is when $F(x_k, w_{k+1}) = \bar{F}(x_k) + w_{k+1}$, where $\{w_i\}_{i \geq 1}$ are *i.i.d.* subgaussian random variables. Moreover, note that this assumption only needs to be satisfied at the fixed points $x^*$, and not for every $x \in \mathbb{R}^d$.

Next, we state the main result of our paper.

**Theorem 4.1.** *Fix $k \geq 1$ and $\xi < 1$. Consider the SA 1.1, and suppose that for any $\delta' \in (0, 1)$, with probability at least $1 - \delta'$, we have $\|x_i - x^*\|_c^2 \leq \alpha_i f_\xi(\delta', k)$ for all $0 \leq i \leq k$, for some function $f_\xi$. Then, under assumptions 4.1, 4.2, and 4.3, for all $\delta \in (0, 1)$, with probability at least $1 - \delta$, we have*

$$\|y_k - x^*\|_c^2 \leq \left(\sqrt{\tilde{\epsilon}\left(k, \frac{\delta}{2}\right)} + \sqrt{\bar{\epsilon}\left(k, \frac{\delta}{2}\right)}\right)^2,$$

*where*

$$\tilde{\epsilon}\left(k, \frac{\delta}{2}\right) = \frac{g(\delta, k)\log\left(\frac{2}{\delta}\right)}{\nu^2(k+1)} + \frac{C_1 d\bar{\sigma}^2}{\nu^2(k+1)} + \frac{C_2 f_\xi\left(\frac{\delta}{2}, k\right)}{(k+1)^{2-\xi}} + \frac{C_3 N^2 \left(f_\xi\left(\frac{\delta}{2}, k\right)\right)^2}{\nu^2(k+1)^{2\xi}} + \frac{C_4 d\hat{\sigma}^2 f_\xi\left(\frac{\delta}{2}, k\right)}{(k+1)^{1+\xi}},$$

$$\bar{\epsilon}\left(k, \frac{\delta}{2}\right) = \frac{C_5 f_\xi(\delta, k_0(\frac{\delta}{2}, k) - 1)\left[(k_0\left(\frac{\delta}{2}, k\right) - 1 + h)^{1-\xi/2} - (h-1)^{1-\xi/2}\right]^2}{(k+1)^2},$$

*with* [2]

$$g(\delta, k) = 24u_{c2}^4 \max\left\{\bar{\sigma}^2, \frac{\alpha(1+M)\hat{\sigma}^2 h^{1-\xi} f_\xi\left(\frac{\delta}{2}, k\right)}{(1-\xi)(k+1)^\xi}\right\},$$

$$k_0\left(\frac{\delta}{2}, k\right) = \min\left\{\left[\left(\frac{\alpha f_\xi\left(\frac{\delta}{2}, k\right)}{R^2}\right)^{1/\xi} - h\right]^+, k+1\right\},$$

*and*

$$C_1 = 3u_{c2}^4, \quad C_2 = \frac{9h^\xi(1+M)}{2\alpha}\left(3 + \frac{\xi^2}{(1-\xi/2)^2}\right), \quad C_3 = \frac{3h^{2-2\xi}\alpha^2(1+M)}{2(1-\xi)^2},$$

$$C_4 = \frac{3h^{1-\xi}u_{c2}^4\alpha(1+M)}{(1-\xi)}, \quad C_5 = \frac{\alpha}{(1-\xi/2)^2}.$$

The proof is given in Appendix B.1. Unlike the case of SA without averaging, where concentration bounds are obtained by establishing a one step recursion [14], here we need to take a more holistic approach and consider all the averaged iterates at once. This involves the added challenge of dealing with sums of noises directly, instead of tackling the noise of each iterate one at a time.

Theorem 4.1 provides a non-asymptotic high-probability bound for the SA iterates (1.1) with averaging. Our bound consists of two terms: $\tilde{\epsilon}(k, \delta/2)$ and $\bar{\epsilon}(k, \delta/2)$. For small enough $k$ such that $k_0(\delta/2, k) = k+1$, the term $\bar{\epsilon}(k, \delta/2)$ decays as $1/k^\xi$, and thus it is the leading term as a function of $k$. For this range of time, the operator $F$ is not assumed to be smooth, and thus our bound decays slower than the expected $1/k$ rate. However, for all $k$ large enough, with high probability, the iterates $x_k$ remain in the neighborhood where $F$ is smooth. Theorem 4.1 states that from that point on, the term $\bar{\epsilon}(k, \delta/2)$ decays as $1/k^2$, and hence it is not a leading term anymore. Furthermore, for the SA with a globally smooth operator, Assumption 4.2(iii) is satisfied with $R = \infty$, and Theorem 4.1 simplifies as follows.

**Corollary 4.1.** *Under assumptions 4.1, 4.2 with $R = \infty$, and 4.3, for all $\delta \in (0, 1)$, with probability at least $1 - \delta$, we have $\|y_k - x^*\|_c^2 \leq \tilde{\epsilon}(k, \delta)$.*

In the following subsections, we explore various implications of our main result.

## 4.1 Tail of the Leading and Higher Order Terms

We first look at the tail behavior in our bound. Throughout this discussion we assume that $f_\xi(\delta, \cdot)$ is non-increasing in $\delta$ and $f_\xi(\delta, \cdot) \in \Omega(\log(1/\delta))$, and that $f_\xi(\cdot, k) \in \mathcal{O}(\text{polylog}(k))$. Note that this assumption is reasonable as the rate of convergence of the error $\|x_k - x^*\|_c^2$ is typically $\tilde{\mathcal{O}}(\alpha_k)$, and the tail of the error $\|x_k - x^*\|_c^2$ is typically sub-exponential or heavier [14]. In this case, our result implies that, if $\xi > 1/2$ or $N = 0$, then the error of the average variable $y_k$ is upper bounded as follows

$$\|y_k - x^*\|_c^2 \leq \underbrace{\mathcal{O}(1/k)\mathcal{O}(\log(1/\delta))}_{\text{leading term}} + \underbrace{o(1/k).\mathcal{O}\left(f_\xi(\delta, k)^2\right)}_{\text{higher order term}}.$$

Note that the leading term is sub-exponential, and the higher order term has a tail that is potentially heavier than exponential, depending on $f_\xi$. In particular, this implies that there exists a sequence of random variables $\{Z_k\}_{k \geq 0}$ such that $\sqrt{k}\|y_k - x^*\| \leq Z_k$ for all $k \geq 0$, and such that $Z_k$ converges to a subgaussian as $k \to \infty$, although for every $k < \infty$, the error $\sqrt{k}\|y_k - x^*\|$ could be heavier than subgaussian. Specifically, we have the following proposition.

**Proposition 4.1.** *There exists an operator $F$ and a sequence of random variables $\{w_i\}_{i \geq 1}$, which satisfy assumptions 4.2, and 4.3, such that $\sqrt{k}\|y_k - x^*\|$ has a heavier tail than any Gaussian for all $k \geq 2$, while*

$$\limsup_{k \to \infty} \mathbb{P}\left(\sqrt{k}\|y_k - x^*\|_c \geq \epsilon\right) \leq a(1 - \Phi(b\epsilon)),$$

*where $a, b > 0$, and $\Phi(\cdot)$ is the CDF of the standard normal distribution.*

---

[2]Given two norms $\|\cdot\|_a$ and $\|\cdot\|_b$, we define the constants $\ell_{ab}$ and $u_{ab}$ such that $\ell_{ab}\|\cdot\|_b \leq \|\cdot\|_a \leq u_{ab}\|\cdot\|_b$.

This is consistent with CLT-style results on the limiting distribution of the error $y_k - x^*$. However, this also suggests that Gaussian can be a poor approximation for the distribution of $y_k - x^*$ for finite $k$, as the distribution of the error could even be heavy-tailed.

Note that in Theorem 4.1, while we assume that the distribution of the squared error $\|x_k - x^*\|_c^2$ is upper bounded by $f_\xi$, the distribution of the squared error of the average $\|y_k - x^*\|_c^2$ is bounded by $f_\xi^2$, which is heavier. Since the average of random variables cannot have a heavier tail than the original random variables, this must be an artifact of our proof. However, one can easily establish a high-probability bound which, for every finite $k$, the bound on the distribution of the squared error of the average has the same tail as the non-averaged iterates. Such a bound is given in the following lemma.

**Lemma 4.1.** *Fix $k \geq 1$. Assume that for any $\delta' \in (0, 1)$ with probability at least $1 - \delta'$, we have $\|x_i - x^*\|_c^2 \leq \alpha_i f_\xi(\delta', k)$ for all $0 \leq i \leq k$. Then with probability at least $1 - \delta$, we have*

$$\|y_k - x^*\|_c^2 \leq \frac{\alpha h^{2-\xi}}{(1 - \xi/2)^2} \frac{f(\delta, k+1)}{(k+1)^\xi}.$$

This result implies that that $\|y_k - x^*\|_c^2$ has at worst the same tail as $\|x_k - x^*\|_c^2$. However, such tighter tail comes at the expense of a sup-optimal convergence rate of $\mathcal{O}(1/k^\xi)$ instead of $\mathcal{O}(1/k)$. Taking the minimum of the result in Theorem 4.1 and Lemma 4.1 we get the best of both worlds, as in the following corollary.

**Corollary 4.2.** *Under the same assumptions as Theorem 4.1, with probability at least $1 - \delta$, we have*

$$\|y_k - x^*\|_c^2 \leq \min \left\{ \mathcal{O}\left(\frac{1}{k}\right) \mathcal{O}\left(\log(1/\delta)\right) + \mathcal{O}\left(\frac{1}{k^{\min\{2\xi, 2-\xi\}}}\right) \tilde{\mathcal{O}}\left(f_\xi(\delta, 0)^2\right), \mathcal{O}\left(\frac{1}{k^\xi}\right) \tilde{\mathcal{O}}(f_\xi(\delta, 0)) \right\}.$$

Corollary 4.2 shows that for any fixed time step $k$, as $\delta \to 0$, we have $\|y_k - x^*\|_c^2 \in \mathcal{O}(f_\xi(\delta, 0))$, while for a fixed $\delta$ as $k \to \infty$, we have $\|y_k - x^*\|_c^2 \in \mathcal{O}(1/k)$. This shows that employing averaging will improve the convergence rate in $k$, while maintaining the same high-probability tail bound as the original iterates. We defer a more detailed comparison of averaged SA vs pure SA to Section 5.

## 4.2 Tightness of the Leading Term

Next, we discuss how tight the leading term in $\tilde{\epsilon}(k, \delta)$ is. For that, we construct an example of a SA where the error is only a constant factor away from our leading term.

Suppose that the dimension $d$ is even. Consider a 2-dimensional i.i.d. sequence of random variables $\{z_i\}_{i \geq 1}$ such that $z_i \sim \mathcal{N}(0, (\bar{\sigma}^2/d)I)$. Using these, we construct a $d$-dimensional sequence of i.i.d. random variables $\{w_i\}_{i \geq 0}$ such that the $j$'th element of $w_i$ is equal to the first element of $z_i$ when $j$ odd, i.e., $w_{i,j} = z_{i,1}$, and it is equal to the second element of $z_i$ when $j$ even is even, i.e., we have $w_{i,j} = z_{i,2}$. Consider the operator $F(x, w) = w$, and the step sizes $\alpha_k = \alpha$ for all $k \geq 0$. This setting satisfies assumptions 4.2 and 4.3 for $\|\cdot\|_c = \|\cdot\|_2$, with $\nu = 1$ and $R = \infty$. In addition, we have the unique fixed point $x^* = 0$. The following proposition bounds the error for this case.

**Proposition 4.2.** *For the example above, for all even dimensions $d$, and for any $\delta \in (0, 1)$, we have*

$$\mathbb{P}\left(\frac{\bar{\sigma}\sqrt{k \log(1/\delta)} - \|x_0\|_2}{k+1} \geq \|y_k - x^*\|_2\right) \leq 1 - \delta.$$

Recall that for large $k$, small $\delta$, and $\|\cdot\|_c = \|\cdot\|_2$, the leading term in Theorem 4.1 is $2\sqrt{6}\bar{\sigma}/\sqrt{k+1}$. Proposition 4.2 implies that that this upper bound is at most a factor of $2\sqrt{6}$ away from the error for this example. Thus, Theorem 4.1 is at most this universal constant away from the tightest possible bound.

## 4.3 Linear versus Non-Linear Operators

We now discuss how our bound depends on the linearity or non-linearity of the operators, and on whether the noise is additive or not. In particular, this has an important effect on what choice of $\xi$ maximizes the convergence rate of the higher order terms in our upper bound.

1. **Linear Operator + Additive Noise:** Suppose that $F(x_k, w_k) = Ax_k + w_k$. It is easy to see that Assumption 4.2 is satisfied with $N = 0$ and that Assumption 4.3 is satisfied with $\hat{\sigma} = 0$. Hence, with probability at least $1 - \delta$, we have

$$\|y_k - x^*\|_c^2 \leq \mathcal{O}(1/k)\mathcal{O}(\log(1/\delta)) + \mathcal{O}\left(\frac{1}{k^{2-\xi}}\right)\tilde{\mathcal{O}}\big(f_\xi(\delta, 0)\big).$$

In this case a smaller $\xi$ achieves a better convergence rate for the higher order terms. Furthermore, the tail of the higher order term is the same as the tail of the errors $\|x_i - x^*\|^2$ for $i \geq 0$, which could be heavier than subgaussian.

2. **Linear Operator + Non Additive Noise:** Suppose that $N = 0$ and $\hat{\sigma}^2 > 0$. Then, we have

$$\|y_k - x^*\|_c^2 \leq \mathcal{O}(1/k)\mathcal{O}(\log(1/\delta)) + \mathcal{O}\left(\frac{1}{k^{1+\xi}} + \frac{1}{k^{2-\xi}}\right).\tilde{\mathcal{O}}\big(\log(1/\delta)f_\xi(\delta, 0)\big).$$

Here the best choice of $\xi$ is $1/2$, and the tail of the higher order term is heavier than the error $\|x_i - x^*\|^2$ by a factor of $\log(1/\delta)$. For instance, if $\|x_i - x^*\|_c^2$ is Weibull with shape parameter $m$, then the higher order term is sub-Weibull with shape parameter $m + 1$.

3. **Nonlinear Operator:** If $N > 0$, we have

$$\|y_k - x^*\|_c^2 \leq \mathcal{O}(1/k)\mathcal{O}(\log(1/\delta)) + \mathcal{O}\left(\frac{1}{k^{2\xi}} + \frac{1}{k^{2-\xi}}\right).\tilde{\mathcal{O}}\big(f_\xi(\delta, 0)^2\big).$$

Our result suggest that the best choice of $\xi$ is $2/3$. In this case, the tail of the higher order term is twice as heavy as the tail of $\|x_i - x^*\|_c^2$.

In general, we observe that for nonlinear operators, we get the same rate of convergence for higher-order terms, regardless of whether the noise is additive or not, and this rate is worse than in the case of linear operators.

*Remark.* For the case of constant step sizes, i.e., for $\xi = 0$, our high-probability bound is of order $\mathcal{O}(N^2\alpha^2) + o(1)$ when the operator is non-linear. This is in line with [27] where they show that the bias of a SA with constant step size $\alpha$ and a non-linear operator is proportional to $\alpha$. On the other hand, if the operator is linear, we have $N = 0$, and the bias disappears. This is consistent with [36].

## 5 Application to Contractive Operators

In this section we will apply our results for the important class of contractive operators $\bar{F}$, which is specified as follows.

**Assumption 5.1.** There exists a constant $\gamma_c \in [0, 1)$ and a norm $\|\cdot\|_c$ such that

$$\|\bar{F}(x_1) - \bar{F}(x_2)\|_c \leq \gamma_c\|x_1 - x_2\|_c, \qquad \forall x_1, x_2 \in \mathbb{R}^d.$$

Recall that in Theorem 4.1 we require a high-probability bound on the SA iterates $\{x_k\}_{k \geq 0}$. To obtain such a high-probability bound, we use the prior work [14] that differentiates between two cases, additive and multiplicative noise, which will be studied in sections 5.1 and 5.2, respectively.

### 5.1 Additive Noise

We first study the additive noise setting (as it was defined in [14]), which is specified in the following assumption.

**Assumption 5.2.** The random vector $F(x_k, w_{k+1}) - \bar{F}(x_k)$ is subgaussian, i.e., there exist $\sigma > 0$ and a (possibly dimension-dependent) constant $c_d > 0$ such that for any $k \geq 0$ and $\mathcal{F}_k$-measurable random vector $v$, the following two inequalities hold:

$$\mathbb{E}\left[\exp\left(\lambda\langle F(x_k, w_{k+1}) - \bar{F}(x_k), v\rangle\right)\big|\mathcal{F}_k\right] \leq \exp\left(\frac{\lambda^2\sigma^2(\|v\|_c^*)^2}{2}\right), \qquad \forall \lambda > 0, \qquad (5.1)$$

where $\|\cdot\|_c^*$ is the dual of the norm $\|\cdot\|_c$, and

$$\mathbb{E}\left[\exp\left(\lambda\|F(x_k, w_{k+1}) - \bar{F}(x_k)\|_c^2\right)\big|\mathcal{F}_k\right] \leq \left(1 - 2\lambda\sigma^2\right)^{-\frac{c_d}{2}}, \qquad \forall \lambda \in \left(0, 1/2\sigma^2\right). \quad (5.2)$$

Employing [14, Theorem 2.4] in Theorem 4.1, we get the following result.

**Proposition 5.1.** *Under assumptions 4.1, 4.2, 4.3, 5.1, and 5.2, using $\xi < 1$ and $h \geq \left(\frac{2\xi}{(1-\gamma_c)\alpha}\right)^{1/(1-\xi)}$, with probability at least $1 - \delta$, we have*

$$\|y_k - x^*\|_c^2 \leq \frac{24\bar{\sigma}^2 \log(2/\delta) + 3du_{c2}^2\bar{\sigma}^2}{(1 - \gamma_c)^2 (k + 1)} + \mathcal{O}(M)\tilde{\mathcal{O}}\left(\frac{1}{k^{2-\xi}} + \frac{1}{k^{1+\xi}}\right)\mathcal{O}\left(\log(1/\delta)\right)$$

$$+ \mathcal{O}(MN^2)\tilde{\mathcal{O}}\left(\frac{1}{k^{2\xi}}\right)\mathcal{O}\left(\log^2(1/\delta)\right). \tag{5.3}$$

Next, we aim at comparing the high-probability bound achieved by averaging vs the high-probability bound achieved through Theorem 2.4 in [14] with $\xi = 1$. For the sake of comparison, we assume that $\|\cdot\|_c = \|\cdot\|_2$. In that case we have $u_{c2} = 1$ and $M = 1$. By Proposition 5.1, with probability at least $1 - \delta$, we have (up to the leading term)

$$\|y_k - x^*\|_c^2 \lessapprox \frac{24 \log(1/\delta) + 3(d + 8 \log(2))}{(k + 1)} \frac{\bar{\sigma}^2}{(1 - \gamma_c)^2}.$$

Moreover, applying [14, Theorem 2.4] with $\xi = 1$ and $k = K$, we obtain

$$\|x_k - x^*\|_c^2 \lessapprox \frac{32a \log(1/\delta) + 32a^2 ec_d(1 + \mu)/(a - 1)}{(k + h)} \frac{\sigma^2}{(1 - \gamma_c)^2},$$

for any $\mu > 0$ and $a > 1$ (refer to Appendix A for calculations). Here, $\bar{\sigma}^2$ is the variance of the subgaussian noise only at $x^*$, while $\sigma^2$ is a uniform bound on the variance over all $x \in \mathbb{R}^d$. Hence, we have $\bar{\sigma}^2 \leq \sigma^2$.

Our results suggest that for SA with additive noise, averaging could improve the constant in the leading term. In particular, in the first term, SA has $32a\sigma^2$ which is at least $32\sigma^2$. Averaging improves this term to $24\bar{\sigma}^2$. In the second term SA has $32\frac{a^2}{a-1}ec_d(1 + \mu)\sigma^2$ which is at least $128ec_d\sigma^2$. For the case that $c_d \approx d$ (which happens when the noise is $\mathcal{N}(0, I_{d\times d})$), averaging improves this term to $3(d + 8\log(2))\bar{\sigma}^2$. We will do a more specific comparison of SA with and without averaging for RL algorithms in Section 6.

*Remark.* It can be shown that, even if we use a smaller value of $h$ than the lower bound $\left(\frac{2\xi}{(1-\gamma_c)\alpha}\right)^{1/(1-\xi)}$ in Proposition 5.1, a concentration bound similar to the one of Equation (5.3) still holds, albeit with larger constants in the higher-order terms.

## 5.2 Multiplicative Noise

Next, we examine the multiplicative noise scenario (as it was defined in [14]), which is characterized by the following assumption:

**Assumption 5.3** ([14])**.** There exists $\sigma > 0$ such that

$$\|F(x_k, w_{k+1}) - \bar{F}(x_k)\|_c \leq \sigma(1 + \|x_k\|_c), \qquad \forall k \geq 0,$$

where $\|\cdot\|_c$ is the norm from Assumption 5.1.

Utilizing [12, Corollary 2.2], Markov's inequality, and Theorem 4.1, we derive the following result:

**Theorem 5.1.** *Under assumptions 4.1, 4.2, 4.3, 5.1, 5.3, and $\xi > 1/2$, for any $\delta \in (0, 1)$, with probability at least $1 - \delta$, we have*

$$\|y_k - x^*\|^2 \leq c\frac{\log(1/\delta) + d}{k + 1} + \mathcal{O}\left(\frac{N}{k^{2\xi}}\right)\mathcal{O}\left(\frac{1}{\delta^2}\right) + \left(\frac{1}{k^{\min\{1+\xi, 2-\xi\}}}\right)\mathcal{O}\left(\frac{1}{\delta}\right).$$

Theorem 5.1 provides a heavy-tailed upper bound for the error. Although this is merely an upper bound, the actual distribution of the error is indeed heavy-tailed, as demonstrated by the following impossibility result:

**Theorem 5.2.** *Under assumptions 5.1 and 5.3, when $\xi \in (0,1)$, there does not exist any $c_0 > 0$ and $m > 0$, such that for all $\delta \in (0,1)$ with probability at least $1 - \delta$*

$$\|y_k - x^*\|_c^2 \leq c_0 \frac{1 + (\log(1/\delta))^m}{k+1}.$$

The above result highlights a fundamental difference between multiplicative and additive noise scenarios. Specifically, while Theorem 2.1 from [14] shows that simple SA with step size $\alpha_k = \alpha/(k+h)$ attains a sub-Weibull distribution, our Theorem 5.2 demonstrates that averaging applied to general SA under multiplicative noise yields a heavier tail than any sub-Weibull distribution. This contrasts sharply with the additive noise scenario, where averaging maintains a subexponential tail similar to standard SA, while improving the constant factor of the leading term.

## 6 Application to Reinforcement Learning

In this section, we study the application of our result to TD-learning and $Q$-learning algorithms.

**TD-Learning:** Consider a finite Markov Decision Process characterized by state space $\mathcal{S}$, action space $\mathcal{A}$, reward function $\mathcal{R}(s,a) : \mathcal{S} \times \mathcal{A} \to [0, R_{\max}]$, and transition probability matrix $P$. We assume $|\mathcal{S}| \geq 2$. The objective of TD-learning is to estimate the value function associated with a policy $\pi$, defined as $V^\pi(s) = \mathbb{E}[\sum_{i=0}^{\infty} \gamma^i \mathcal{R}(S_i, A_i)|S_0 = s]$. We focus on asynchronous $TD(n)$ with i.i.d. noise, given by

$$V_{k+1}(s) = V_k(s) + \alpha_k \mathbb{1}_{\{s=S_k^0\}} \left( \sum_{i=k}^{k+n-1} \gamma^{i-k} \Big( \mathcal{R}(S_k^i, A_k^i) + \gamma V_k(S_k^{i+1}) - V_k(S_k^i) \Big) \right), \quad (6.1)$$

for all $s \in \mathcal{S}$. Here, $S_0^0, S_1^0, \ldots \overset{iid}{\sim} \mu^\pi$, where $\mu^\pi$ denotes the stationary distribution of the induced Markov chain under policy $\pi$. For each $k$, $\{(S_k^i, A_k^i)\}_{0 \leq i \leq n}$ represents a trajectory following policy $\pi$, with transitions $A_k^i \sim \pi(\cdot|S_k^i)$, and $S_k^{i+1} \sim P(\cdot|S_k^i, A_k^i)$. We assume that the initial point of the algorithm satisfies $V_0(s) \in [0, \frac{R_{\max}}{1-\gamma}]$ for all $s \in \mathcal{S}$ and that $\alpha_k = \alpha/\sqrt{k+h}$ with $\alpha/\sqrt{h} \leq 1$. We denote $\bar{V}_k = \frac{1}{k+1} \sum_{i=0}^{k} V_i$.

**Theorem 6.1.** *Under the asynchronous $TD(n)$ algorithm (6.1), for any $\delta \in (0,1)$, with probability at least $1 - \delta$, the following bound holds:*

$$\|\bar{V}_k - V^\pi\|_\infty^2 \leq \frac{R_{\max}^2}{(1-\gamma)^2(1-\gamma^n)^2(\mu_{\min}^\pi)^2} \frac{24 \log(2/\delta) + 3d}{(k+1)} + \tilde{\mathcal{O}}\left(\frac{1}{k^{5/4}}\right) \mathcal{O}(\log(1/\delta)).$$

To the best of our knowledge, Theorem 6.1 establishes the first bound on the entire distribution of the error for averaged TD-learning. In particular, prior works such as [19, 49, 36] provide bounds for the raw iterates or use step sizes that depend on a fixed confidence level $\delta$, and thus only give pointwise control (i.e., one quantile of the error distribution). In contrast, since our step size is independent of $\delta$, we obtain bounds that control the entire tail of the error distribution for the averaged iterates.

**Q-Learning:** Next, we study the asynchronous $Q$-learning algorithm with i.i.d. noise, given by

$$Q_{k+1}(s,a) = Q_k(s,a) + \alpha_k \mathbb{1}_{\{(s,a)=(S_k, A_k)\}} \left( \mathcal{R}(S_k, A_k) + \gamma \max_{a'}\{Q_k(S_k', a')\} - Q_k(S_k, A_k) \right),$$

for all $(s,a) \in \mathcal{S} \times \mathcal{A}$, where $S_k \sim \mu^{\pi_b}, A_k \sim \pi_b(\cdot|S_k), S_k' \sim P(\cdot|S_k, A_k)$ and $\pi_b$ is some fixed sampling policy. We denote $\rho_b = \min_{s,a} \mu^{\pi_b}(s)\pi_b(a|s)$. We assume that the initial point of the algorithm satisfies $Q_0(s,a) \in [0, \frac{R_{\max}}{1-\gamma}]$ for all $(s,a) \in \mathcal{S} \times \mathcal{A}$ and $\alpha_k = \alpha/\sqrt{k+h}$ with $\alpha/\sqrt{h} < 1$. Without loss of generality, in this section we also assume $|\mathcal{A}| \geq 2$. We denote the optimal $Q$-function as $Q^*$ and $\bar{Q}_k = \frac{1}{k+1} \sum_{i=0}^{k} Q_i$. We further impose the following common [56] assumption on $Q^*$.

**Assumption 6.1.** $Q^*$ is greedily unique, i.e. for every $s \in \mathcal{S}$, $a^*(s) = \arg\max_{a'}\{Q^*(s,a')\}$ is unique.

**Theorem 6.2.** *For asynchronous Q-learning with i.i.d. noise and $\xi = 1/2$, under Assumption 6.1, for any $\delta \in (0, 1)$, with probability at least $1 - \delta$, we have*

$$\|\bar{Q}_k - Q^*\|_\infty^2 \leq \frac{12R_{\max}^2}{(1-\gamma)^4\rho_b^2} \frac{8 \log(2/\delta) + |\mathcal{S}||\mathcal{A}|}{(k+1)} + \tilde{\mathcal{O}}\left(\frac{1}{k^{5/4}}\right)\mathcal{O}(\log(1/\delta)).$$

To the best of our knowledge, Theorem 6.2 establishes the first bound on the entire distribution of the error for averaged $Q$-learning.

We observe that, since the noise structure in tabular TD-learning and $Q$-learning is the same (both corresponding to additive noise), the bounds are of the same order of $k$ and $\delta$.

**Off-policy TD-learning:** Theorems 6.1 and 6.2 are two instances of RL algorithms which can be modeled with SA with additive noise, as in both of these settings the noise is bounded. Next, we study off-policy TD-learning with linear function approximation which has multiplicative noise. In this setting we assume we have access to a matrix $\Phi \in \mathbb{R}^{|\mathcal{S}| \times d}$, and we denote the $s$-th row of this matrix by $\phi(s)$. The goal is to find $v^\pi \in \mathbb{R}^d$ that estimates the value function as $V^\pi(s) \approx v^\pi\phi(s)$ through the solution of the following fixed point equation $\Phi v^\pi = \Pi_\Phi^{\pi_b}((\mathcal{T}^\pi)^n\Phi v^\pi)$ Here, $\Pi_\Phi^{\pi_b} = \Phi(\Phi^\top\mathcal{K}^{\pi_b}\Phi)^{-1}\Phi^\top\mathcal{K}^{\pi_b}$ is a linear function that projects into the subspace spanned by the matrix $\Phi$, and $\mathcal{K}^{\pi_b}$ is a diagonal matrix, with diagonal entries equal to the stationary distribution of the behavior policy $\pi_b$. Furthermore, $\mathcal{T}^\pi$ is the Bellman operator. We employ TD($n$) as follows:

$$v_{k+1} = v_k + \alpha_k\phi(S_k^0)\sum_{l=0}^{n-1}\gamma^l\left[\prod_{j=0}^l\frac{\pi(A_k^j|S_k^j)}{\pi_b(A_k^j|S_k^j)}\right]\left[\mathcal{R}(S_k^l, A_k^l) + \gamma\phi(S_k^{l+1})^\top v_k - \phi(S_k^l)^\top v_k\right],$$

where $S_0^0, S_1^0, \ldots \overset{iid}{\sim} \mu^{\pi_b}$, and $\mu^{\pi_b}$ is the stationary distribution over states of the induced Markov chain by following policy $\pi_b$, and for every $k$, $\{(S_k^i, A_k^i)\}_{0 \leq i \leq n}$ is a single trajectory of state-action pairs following policy $\pi_b$ as $A_k^i \sim \pi_b(\cdot|S_k^i)$, $S_k^{i+1} \sim P(\cdot|S_k^i, A_k^i)$, $0 \leq i \leq n$. We assume that $\alpha_k = \alpha/(k+h)^\xi$. We denote $\bar{v}_k = \frac{1}{k+1}\sum_{i=0}^k v_i$.

**Theorem 6.3.** *For large enough $n$, there exists a constant $c > 0$ such that, for any $\delta \in (0, 1)$, with probability at least $1 - \delta$, we have*

$$\|v_k - v^\pi\|_2^2 \leq c\frac{\log(1/\delta) + d}{k+1} + \mathcal{O}\left(\frac{1}{k^{\min\{1+\xi, 2-\xi\}}}\right)\mathcal{O}\left(\frac{1}{\delta}\right).$$

To the best of our knowledge, Theorem 6.3 establishes the first bound on the entire distribution of the error for the averaged off-policy TD($n$) with linear function approximation.

Note that for TD-learning with linear function approximation, the tail has $\mathcal{O}(1/\delta)$ dependence, which is a sub-Pareto distribution. Moreover, TD-learning with linear function approximation is an example of SA with multiplicative noise, and as shown in Theorem 5.2, the tail has $\Omega((\log(1/\delta))^n)$ for all $n \geq 1$. In particular, this means that the tail of TD-learning with linear function approximation is heavier than any sub-weibull, but it is at most as heavy as sub-pareto. In contrast, as shown in Theorems 6.1 and 6.2, tabular TD-learning and $Q$-learning have $\mathcal{O}(\log(1/\delta))$ tail, which is sub-exponential.

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

# A   Analysis of the case study in Section 5

The generalized Moreau envelope norm $\|\cdot\|_m$ was introduced in [14], and it is defined as

$$\|x\|_m^2 = \inf_{u \in \mathbb{R}^d} \left\{ \|u\|_c^2 + \frac{1}{\mu}\|x - u\|_s^2 \right\}.$$

By choosing $\|\cdot\|_s = \|\cdot\|_c$, as shown in [14], $\frac{1}{2}\|\cdot\|_m^2$ is $M$-smooth. Furthermore, we have $\|\cdot\|_m = \|\cdot\|_c/\sqrt{1+\mu}$. Hence, $u_{cm} = \ell_{cm} = \sqrt{1+\mu}$. Also, we choose $\alpha = a/(1 - \gamma_c)$ with $a > 1$. Note that $u_{mc*} = u_{mc}u_{cc*} = u_{cc*}/\sqrt{1+\mu}$. Moreover, the contraction factor under the norm $\|\cdot\|_m$ is $\gamma_c$.

# B   Proofs of main results

## B.1   Proof of Theorem 4.1

For ease of notation, we drop $\xi$ from $f_\xi(\delta, k)$.

We define the event

$$E_k(\delta) := \{\|x_i - x^*\|_c^2 \le \alpha_i f(\delta, k) \ \text{ for all } \ 0 \le i \le k\},$$

which we assumed to have $\mathbb{P}(E_k) \ge 1 - \delta$. For ease of notation, we use $E_k$ instead of $E_k(\delta)$.

In order to obtain a tight high probability bound, we need to exploit Assumption 4.2(iii), which only holds whenever $\|x_k - x^*\|_c \le R$. We can ensure that with high probability, the SA is within the local smooth range of the operator after a certain time. Define $\tilde{k}_0(\delta, k) \ge 0$ to be the smallest number such that $\alpha_{\tilde{k}_0(\delta,k)} f(\delta, k) \le R^2$. Solving for the smallest $\tilde{k}_0(\delta, k)$ we have

$$\tilde{k}_0(\delta, k) = \max \left\{ 0, \left\lceil \left( \frac{\alpha f(\delta, k)}{R^2} \right)^{1/\xi} - h \right\rceil \right\}.$$

We define $k_0(\delta, k) := \min\left\{ \tilde{k}_0(\delta, k), k + 1 \right\}$. For ease of notation, unless otherwise stated, we denote $k_0(\delta, k)$ by $k_0$.

For any $k \ge 0$, we have

$$y_k = \frac{1}{k+1} \sum_{i=0}^{k} x_i$$

$$= \frac{1}{k+1} \sum_{i=0}^{k_0-1} x_i + \frac{1}{k+1} \sum_{i=k_0}^{k} x_i.$$

We define

$$\bar{y}_k := \frac{1}{k+1} \sum_{i=0}^{k_0-1} (x_i - x^*) \qquad \text{and} \qquad \tilde{y}_k = \frac{1}{k+1} \sum_{i=k_0}^{k} (x_i - x^*).$$

It is clear that $y_k - x^* = \bar{y}_k + \tilde{y}_k$. Applying norm on both sides, and using triangle inequality, we get

$$\|y_k - x^*\|_c \le \|\bar{y}_k\|_c + \|\tilde{y}_k\|_c. \tag{B.1}$$

We study $\|\bar{y}_k\|_c$ as follows. For every sample path in the event $E_{k_0}$, we have

$$\|\bar{y}_k\|_c = \left\| \frac{1}{k+1} \sum_{i=0}^{k_0-1} x_i - x^* \right\|_c$$

$$\le \frac{1}{k+1} \sum_{i=0}^{k_0-1} \|x_i - x^*\|_c \qquad \text{(triangle inequality)}$$

$$\leq \frac{\sqrt{f(\delta, k_0)}}{k+1} \sum_{i=0}^{k_0-1} \sqrt{\alpha_i} \qquad\qquad (\|x_i - x^*\|_c \leq \sqrt{\alpha_i f_\xi(\delta', k)} \text{ in } E_{k_0})$$

$$\leq \frac{\sqrt{\alpha f(\delta, k_0)}}{k+1} \int_{-1}^{k_0-1} \frac{1}{(x+h)^{\xi/2}} dx$$

$$= \frac{\sqrt{\alpha f(\delta, k_0)}[(k_0 - 1 + h)^{1-\xi/2} - (h-1)^{1-\xi/2}]}{(1 - \xi/2)(k+1)}.$$

Hence, for any $k \geq 0$, with probability at least $1 - \delta$ we have

$$\|\bar{y}_k\|_c^2 \leq \frac{\alpha f(\delta, k_0) \left[(k_0 - 1 + h)^{1-\xi/2} - (h-1)^{1-\xi/2}\right]^2}{(1 - \xi/2)^2 (k+1)^2} =: \bar{\epsilon}(k, \delta). \tag{B.2}$$

Next, we study $\|\tilde{y}_k\|_c$. We have

$$(J_{\bar{F}}(x^*) - I)(x_k - x^*) = (F(x_k, w_{k+1}) - x_k) - (F(x^*, w_{k+1}) - x^*)$$
$$+ [(J_{F_w}(x^*, w_{k+1}) - I)(x_k - x^*) - (F(x_k, w_{k+1}) - x_k) + (F(x^*, w_{k+1}) - x^*)]$$
$$+ [J_{\bar{F}}(x^*) - J_{F_w}(x^*, w_{k+1})](x_k - x^*).$$

Hence,

$$(k+1)(J_{\bar{F}}(x^*) - I)(\tilde{y}_k) = \sum_{i=k_0}^{k} [(F(x_i, w_{i+1}) - x_i)]$$
$$- \sum_{i=k_0}^{k} (F(x^*, w_{i+1}) - x^*)$$
$$+ \sum_{i=k_0}^{k} [(J_{F_w}(x^*, w_{i+1}) - I)(x_i - x^*) - (F(x_i, w_{i+1}) - x_i) + (F(x^*, w_{i+1}) - x^*)]$$
$$+ \sum_{i=k_0}^{k} [J_{\bar{F}}(x^*) - J_{F_w}(x^*, w_{i+1})](x_i - x^*)$$
$$= \sum_{i=k_0}^{k} \left[\frac{x_{i+1} - x_i}{\alpha_i}\right]$$
$$- \sum_{i=k_0}^{k} (F(x^*, w_{i+1}) - x^*)$$
$$+ \sum_{i=k_0}^{k} [(J_{F_w}(x^*, w_{i+1}) - I)(x_i - x^*) - (F(x_i, w_{i+1}) - x_i) + (F(x^*, w_{i+1}) - x^*)]$$
$$+ \sum_{i=k_0}^{k} [J_{\bar{F}}(x^*) - J_{F_w}(x^*, w_{i+1})](x_i - x^*). \tag{B.3}$$

By Assumption 4.1, we have

$$\|a + b\|_c^2 \leq \|a\|_c^2 + \langle \nabla \|a\|_c^2, b \rangle + \frac{M}{2} \|b\|_c^2$$

$$\leq \|a\|_c^2 + \left\|\nabla \|a\|_c^2\right\|_{c*} \|b\|_c + \frac{M}{2} \|b\|_c^2 \qquad \text{(Hölder's inequality)}$$

$$\leq \|a\|_c^2 + \|a\|_c \|b\|_c + \frac{M}{2} \|b\|_c^2 \qquad \text{([47, Lemma 2.6])}$$

$$\leq \|a\|_c^2 + \frac{1}{2} \|a\|_c^2 + \frac{1}{2} \|b\|_c^2 + \frac{M}{2} \|b\|_c^2 \qquad \text{(Young's inequality)}$$

$$= \frac{3}{2}\|a\|_c^2 + \frac{1+M}{2}\|b\|_c^2. \tag{B.4}$$

Hence, we have

$$(k+1)^2 \nu^2 \left\| \tilde{y}_k - x^* \right\|_c^2 \mathbb{1}_{E_{k+1}}$$

$$\leq (k+1)^2 \left\| (J_{\bar{F}}(x^*) - I)(\tilde{y}_k - x^*) \right\|_c^2 \mathbb{1}_{E_{k+1}} \qquad \text{(Assumption 4.2(ii))}$$

$$\leq \frac{3}{2} \left\| \sum_{i=k_0}^{k} (F(x^*, w_{i+1}) - x^*) \right\|_c^2 \mathbb{1}_{E_{k+1}}$$

$$+ \frac{1+M}{2} \left\| \sum_{i=k_0}^{k} \left[ \frac{x_{i+1} - x_i}{\alpha_i} \right] \right. $$

$$+ [(J_{F_w}(x^*, w_{i+1}) - I)(x_i - x^*) - (F(x_i, w_{i+1}) - x_i) + (F(x^*, w_{i+1}) - x^*)]$$

$$\left. + [J_{\bar{F}}(x^*) - J_{F_w}(x^*, w_{i+1})](x_i - x^*) \right\|_c^2 \mathbb{1}_{E_{k+1}} \qquad \text{(Eq. (B.3) and (B.4))}$$

$$\leq \frac{3}{2} \left\| \left( \sum_{i=k_0}^{k} (F(x^*, w_{i+1}) - x^*) \right) \right\|_c^2 \mathbb{1}_{E_{k+1}}$$

$$+ \frac{3(1+M)}{2} \left\| \sum_{i=k_0}^{k} \left[ \frac{x_{i+1} - x_i}{\alpha_i} \right] \right\|_c^2 \mathbb{1}_{E_{k+1}}$$

$$+ \frac{3(1+M)}{2} \left\| \sum_{i=k_0}^{k} [(J_{F_w}(x^*, w_{i+1}) - I)(x_i - x^*) \right.$$

$$\left. - (F(x_i, w_{i+1}) - x_i) + (F(x^*, w_{i+1}) - x^*)] \right\|_c^2 \mathbb{1}_{E_{k+1}}$$

$$+ \frac{3(1+M)}{2} \left\| \sum_{i=k_0}^{k} [J_{\bar{F}}(x^*) - J_{F_w}(x^*, w_{i+1})](x_i - x^*) \right\|_c^2 \mathbb{1}_{E_{k+1}}$$
$$\text{(triangle inequality and } (a+b+c)^2 \leq 3a^2 + 3b^2 + 3c^2)$$

$$\leq \frac{3}{2} \left\| \left( \sum_{i=k_0}^{k} (F(x^*, w_{i+1}) - x^*) \right) \right\|_c^2 \mathbb{1}_{E_{k+1}}$$

$$+ \frac{3(1+M)}{2} \left\| \sum_{i=k_0}^{k} \left[ \frac{x_{i+1} - x_i}{\alpha_i} \right] \right\|_c^2 \mathbb{1}_{E_{k+1}}$$

$$+ \frac{3(1+M)}{2} \left( \sum_{i=k_0}^{k} \| [(J_{F_w}(x^*, w_{i+1}) - I)(x_i - x^*) \right.$$

$$\left. - (F(x_i, w_{i+1}) - x_i) + (F(x^*, w_{i+1}) - x^*)] \|_c \right)^2 \mathbb{1}_{E_{k+1}}$$
$$\text{(triangle inequality)}$$

$$+ \frac{3(1+M)}{2} \left\| \sum_{i=k_0}^{k} [J_{\bar{F}}(x^*) - J_{F_w}(x^*, w_{i+1})](x_i - x^*) \right\|_c^2 \mathbb{1}_{E_{k+1}}$$

$$\leq \frac{3}{2} \left\| \left( \sum_{i=k_0}^{k} (F(x^*, w_{i+1}) - x^*) \right) \right\|_c^2 \mathbb{1}_{E_{k+1}}$$

$$+ \frac{3(1+M)}{2} \left\| \sum_{i=k_0}^{k} \left[ \frac{x_{i+1} - x_i}{\alpha_i} \right] \right\|_c^2 \mathbb{1}_{E_{k+1}}$$

$$+ \frac{3(1+M)}{2} N^2 \left( \sum_{i=k_0}^{k} \|x_i - x^*\|_c^2 \right)^2 \mathbb{1}_{E_{k+1}} \qquad \text{(Assumption 4.2(iii))}$$

$$+ \frac{3(1+M)}{2} \left\| \sum_{i=k_0}^{k} [J_{\bar{F}}(x^*) - J_{F_w}(x^*, w_{i+1})](x_i - x^*) \right\|_c^2 \mathbb{1}_{E_{k+1}}.$$

Let us define

$$\lambda_k := \frac{k+1}{24} \min \left\{ \frac{1}{u_{c2}^4 \bar{\sigma}^2}, \frac{(1-\xi)(k+1)^\xi}{u_{c2}^4 \alpha (1+M) \hat{\sigma}^2 f(\delta, k) h^{1-\xi}} \right\}.$$

We have

$$\mathbb{E}[\exp(\lambda_k \nu^2 \|\tilde{y}_k - x^*\|_c^2 . \mathbb{1}_{E_{k+1}})]$$

$$\leq \mathbb{E}\left[ \exp \left( \frac{\lambda_k}{(k+1)^2} \frac{3}{2} \left\| \sum_{i=k_0}^{k} (F(x^*, w_{i+1}) - x^*) \right\|_c^2 . \mathbb{1}_{E_{k+1}} \right) \right.$$

$$. \exp \left( \frac{\lambda_k}{(k+1)^2} \frac{3(1+M)}{2} \left\| \sum_{i=k_0}^{k} \left[ \frac{x_{i+1} - x_i}{\alpha_i} \right] \right\|_c^2 . \mathbb{1}_{E_{k+1}} \right)$$

$$. \exp \left( \frac{\lambda_k}{(k+1)^2} \frac{3(1+M)}{2} N^2 \left( \sum_{i=k_0}^{k} \|x_i - x^*\|_c^2 \right)^2 . \mathbb{1}_{E_{k+1}} \right)$$

$$\left. . \exp \left( \frac{\lambda_k}{(k+1)^2} \frac{3(1+M)}{2} \left\| \sum_{i=k_0}^{k} [J_{\bar{F}}(x^*) - J_{F_w}(x^*, w_{i+1})](x_i - x^*) \right\|_c^2 . \mathbb{1}_{E_{k+1}} \right) \right]$$

$$\leq \underbrace{\left\{ \mathbb{E}\left[ \exp \left( \frac{6\lambda_k}{(k+1)^2} \left\| \sum_{i=k_0}^{k} (F(x^*, w_{i+1}) - x^*) \right\|_c^2 . \mathbb{1}_{E_{k+1}} \right) \right] \right\}^{1/4}}_{T_1}$$

$$. \underbrace{\left\{ \mathbb{E}\left[ \exp \left( \frac{6\lambda_k}{(k+1)^2} (1+M) \left\| \sum_{i=k_0}^{k} \left[ \frac{x_{i+1} - x_i}{\alpha_i} \right] \right\|_c^2 . \mathbb{1}_{E_{k+1}} \right) \right] \right\}^{1/4}}_{T_2}$$

$$. \underbrace{\left\{ \mathbb{E}\left[ \exp \left( \frac{6\lambda_k}{(k+1)^2} (1+M) N^2 \left( \sum_{i=k_0}^{k} \|x_i - x^*\|_c^2 \right)^2 . \mathbb{1}_{E_{k+1}} \right) \right] \right\}^{1/4}}_{T_3}$$

$$. \underbrace{\left\{ \mathbb{E}\left[ \exp \left( \frac{6\lambda_k}{(k+1)^2} (1+M) \left\| \sum_{i=k_0}^{k} [J_{\bar{F}}(x^*) - J_{F_w}(x^*, w_{i+1})](x_i - x^*) \right\|_c^2 . \mathbb{1}_{E_{k+1}} \right) \right] \right\}^{1/4}}_{T_4},$$

(B.5)

where in the last inequality we use Cauchy–Schwarz three times. We study each of the terms above separately.

**The term $T_1$:** By Assumption 4.3 and Lemma C.1, it follows that $\sum_{i=k_0}^{k}(F(x^*, w_{i+1}) - x^*)$ is $(k - k_0 + 1)\bar{\sigma}^2$-subgaussian. Furthermore, by Lemma C.2, we have

$$T_1 = \left\{ \mathbb{E}\left[ \exp\left( \frac{6\lambda_k}{(k+1)^2} \left\| \sum_{i=k_0}^{k}(F(x^*, w_{i+1}) - x^*) \right\|_c^2 . \mathbb{1}_{E_{k+1}} \right) \right] \right\}^{1/4}$$

$$\leq \left\{ \mathbb{E}\left[ \exp\left( \frac{6\lambda_k}{(k+1)^2} u_{c2}^2 \left\| \sum_{i=k_0}^{k}(F(x^*, w_{i+1}) - x^*) \right\|_2^2 . \mathbb{1}_{E_{k+1}} \right) \right] \right\}^{1/4}$$

$$\leq \left\{ \mathbb{E}\left[ \exp\left( \frac{6\lambda_k}{(k+1)^2} u_{c2}^2 \left\| \sum_{i=k_0}^{k}(F(x^*, w_{i+1}) - x^*) \right\|_2^2 \right) \right] \right\}^{1/4}$$

$$\leq \left\{ \frac{1}{\left( 1 - \frac{12\lambda_k u_{c2}^4 \bar{\sigma}^2}{k+1} \right)^{d/2}} \right\}^{1/4}$$

$$= \left( \frac{1}{1 - \frac{12\lambda_k u_{c2}^4 \bar{\sigma}^2}{k+1}} \right)^{d/8}$$

$$\leq \exp\left( \frac{3d\lambda_k u_{c2}^4 \bar{\sigma}^2}{k+1} \right), \qquad \qquad \text{(we use } \lambda_k \leq \frac{k+1}{24 u_{c2}^4 \bar{\sigma}^2}\text{)}$$

where in last inequality we use that $1/(1-x) \leq \exp(2x)$ for all $x \leq 1/2$.

**The term $T_2$:**

$$T_2 = \left\{ \mathbb{E}\left[ \exp\left( \frac{6\lambda_k}{(k+1)^2}(1+M) \left\| \sum_{i=k_0}^{k} \frac{x_i - x^* + x^* - x_{i+1}}{\alpha_i} \right\|_c^2 . \mathbb{1}_{E_{k+1}} \right) \right] \right\}^{1/4}$$

$$\leq \left\{ \mathbb{E}\left[ \exp\left( \frac{6\lambda_k}{(k+1)^2}(1+M) \left\| \frac{x_{k_0} - x^*}{\alpha_{k_0}} - \frac{x_{k+1} - x^*}{\alpha_k} \right. \right. \right. \right.$$

$$\left. \left. \left. \left. + \sum_{i=k_0+1}^{k}(x_i - x^*)\left( \frac{1}{\alpha_{i+1}} - \frac{1}{\alpha_i} \right) \right\|_c^2 . \mathbb{1}_{E_{k+1}} \right) \right] \right\}^{1/4}$$

$$\overset{(a)}{\leq} \left\{ \mathbb{E}\left[ \exp\left( \frac{18\lambda_k}{(k+1)^2}(1+M)\left( \frac{\|x_{k_0} - x^*\|_c^2}{\alpha_{k_0}^2} + \frac{\|x_{k+1} - x^*\|_c^2}{\alpha_k^2} \right. \right. \right. \right.$$

$$\left. \left. \left. \left. + \left\| \sum_{i=k_0+1}^{k}(x_i - x^*)\left( \frac{1}{\alpha_{i+1}} - \frac{1}{\alpha_i} \right) \right\|_c^2 \right) . \mathbb{1}_{E_{k+1}} \right) \right] \right\}^{1/4}$$

$$\overset{(b)}{\leq} \left\{ \mathbb{E}\left[ \exp\left( \frac{18\lambda_k}{(k+1)^2}(1+M)\left( \frac{\|x_{k_0} - x^*\|_c^2}{\alpha_{k_0}^2} + \frac{\|x_{k+1} - x^*\|_c^2}{\alpha_k^2} \right. \right. \right. \right.$$

$$\left. \left. \left. \left. + \left[ \sum_{i=k_0+1}^{k} \|x_i - x^*\|_c \left( \frac{1}{\alpha_{i+1}} - \frac{1}{\alpha_i} \right) \right]^2 \right) . \mathbb{1}_{E_{k+1}} \right) \right] \right\}^{1/4}$$

$$\overset{(c)}{\leq} \exp\left( \frac{9\lambda_k}{2(k+1)^2}(1+M)\left( \frac{f(\delta,k)}{\alpha_{k_0}} + \frac{2f(\delta,k)}{\alpha_k} + f(\delta,k)\left[ \sum_{i=k_0+1}^{k} \sqrt{\alpha_i}\left( \frac{1}{\alpha_{i+1}} - \frac{1}{\alpha_i} \right) \right]^2 \right) \right)$$

$$\overset{(d)}{\leq} \exp\left( \frac{9\lambda_k f(\delta, k)}{2(k+1)^2}(1+M)\left( \frac{1}{\alpha_{k_0}} + \frac{2}{\alpha_k} + \left[ \sum_{i=k_0+1}^{k} \left( \frac{\xi}{\sqrt{\alpha_i}(i+h)} \right) \right]^2 \right) \right)$$

$$= \exp\left( \frac{9\lambda_k f(\delta, k)}{2(k+1)^2}(1+M)\left( \frac{1}{\alpha_{k_0}} + \frac{2}{\alpha_k} + \frac{\xi^2}{\alpha}\left[ \sum_{i=k_0+1}^{k} \frac{1}{(i+h)^{1-\xi/2}} \right]^2 \right) \right)$$

$$\overset{(e)}{\leq} \exp\left( \frac{9\lambda_k f(\delta, k)}{2(k+1)^2}(1+M)\left( \frac{1}{\alpha_{k_0}} + \frac{2}{\alpha_k} + \frac{\xi^2}{\alpha}\left[ \frac{1}{(1-\xi/2)}(k+h)^{\xi/2} \right]^2 \right) \right)$$

$$= \exp\left( \frac{9\lambda_k f(\delta, k)}{2(k+1)^2}(1+M)\left( \frac{1}{\alpha_{k_0}} + \frac{2}{\alpha_k} + \frac{\xi^2}{(1-\xi/2)^2\alpha_k} \right) \right)$$

$$\leq \exp\left( \frac{9\lambda_k f(\delta, k)}{2(k+1)^2\alpha_k}(1+M)\left( 3 + \frac{\xi^2}{(1-\xi/2)^2} \right) \right) \qquad (k \geq k_0)$$

$$\leq \exp\left( \frac{9h^\xi(1+M)\lambda_k f(\delta, k)}{2\alpha(k+1)^{2-\xi}}\left( 3 + \frac{\xi^2}{(1-\xi/2)^2} \right) \right),$$

where $(a)$ and $(b)$ are by triangle inequality and $(a+b+c)^2 \leq 3a^2 + 3b^2 + 3c^2$, $(c)$ is by the assumption of the theorem, $(d)$ is by $(x+1)^\xi - x^\xi \leq \xi/x^{1-\xi}$ for $x > 0$, and $(e)$ is by integral upper bound of summation.

**The term $T_3$:**

$$T_3 = \left\{ \mathbb{E}\left[ \exp\left( \frac{6\lambda_k}{(k+1)^2}(1+M)N^2 \left( \sum_{i=k_0}^{k} \|x_i - x^*\|_c^2 \right)^2 . \mathbb{1}_{E_{k+1}} \right) \right] \right\}^{1/4}$$

$$\leq \left\{ \mathbb{E}\left[ \exp\left( \frac{6\lambda_k}{(k+1)^2}(1+M)N^2 \left( \sum_{i=k_0}^{k} \alpha_i f(\delta, k) \right)^2 \right) \right] \right\}^{1/4}$$

$$(\|x_i - x^*\|_c^2 \leq \alpha_i f_\xi(\delta', k) \text{ in } E_{k+1})$$

$$= \exp\left( \frac{3\lambda_k N^2 (f(\delta, k))^2}{2(k+1)^2}(1+M) \left( \sum_{i=k_0}^{k} \alpha_i \right)^2 \right)$$

$$\leq \exp\left( \frac{3\alpha^2 \lambda_k N^2 (f(\delta, k))^2}{2(k+1)^2}(1+M) \left( \int_{k_0-1}^{k} \frac{1}{(z+h)^\xi} dz \right)^2 \right)$$

$$= \exp\left( \frac{3\alpha^2 \lambda_k N^2 (f(\delta, k))^2}{2(k+1)^2}(1+M) \left( \frac{(k+h)^{1-\xi} - (k_0+h-1)^{1-\xi}}{\xi - 1} \right)^2 \right)$$

$$\leq \exp\left( \frac{3\alpha^2 \lambda_k N^2 (f(\delta, k))^2}{2(k+1)^2}(1+M) \frac{(k+h)^{2-2\xi}}{(\xi-1)^2} \right) \qquad (\text{using } h > 1)$$

$$\leq \exp\left( \frac{9h^{2-2\xi}\alpha^2 \lambda_k N^2(1+M)(f(\delta, k))^2}{2(k+1)^{2\xi}(1-\xi)^2} \right),$$

where in the last inequality we used the fact that $(k+h)^{2-2\xi} \leq (k+1)^{2-2\xi}h^{2-2\xi}$.

**The term $T_4$:** It is easy to see that, almost surely, we have

$$\mathbb{1}_{E_{k+1}} \sum_{i=k_0}^{k} [J_{\bar{F}}(x^*) - J_{F_w}(x^*, w_{i+1})](x_i - x^*) = \mathbb{1}_{E_{k+1}} \sum_{i=k_0}^{k} [J_{\bar{F}}(x^*) - J_{F_w}(x^*, w_{i+1})](x_i - x^*)\mathbb{1}_{E_i}.$$

Hence, we have

$$\left\| \mathbb{1}_{E_{k+1}} \sum_{i=k_0}^{k} [J_{\bar{F}}(x^*) - J_{F_w}(x^*, w_{i+1})](x_i - x^*) \right\|_c = \left\| \mathbb{1}_{E_{k+1}} \sum_{i=k_0}^{k} [J_{\bar{F}}(x^*) - J_{F_w}(x^*, w_{i+1})](x_i - x^*)\mathbb{1}_{E_i} \right\|_c$$

$$= \mathbb{1}_{E_{k+1}} \left\| \sum_{i=k_0}^{k} [J_{\bar{F}}(x^*) - J_{F_w}(x^*, w_{i+1})](x_i - x^*) \mathbb{1}_{E_i} \right\|_c$$

$$\leq \left\| \sum_{i=k_0}^{k} [J_{\bar{F}}(x^*) - J_{F_w}(x^*, w_{i+1})](x_i - x^*) \mathbb{1}_{E_i} \right\|_c .$$

We now show that Assumption 4.3 implies that

$$\mathbb{E}[J_{\bar{F}}(x^*) - J_{F_w}(x^*, w_{k+1}))|\mathcal{F}_k] = 0.$$

Indeed, by Taylor expansion of the inequality (4.2), we get

$$\mathbb{E}[1 + \lambda \langle (J_{\bar{F}}(x^*) - J_{F_w}(x^*, w_{k+1}))a, b \rangle + \mathcal{O}(\lambda^2)|\mathcal{F}_k] \leq 1 + \frac{\lambda^2 \hat{\sigma}^2 \|a\|_c^2 \|b\|_c^2}{2} + \mathcal{O}(\lambda^4).$$

If we had $\mathbb{E}[J_{\bar{F}}(x^*) - J_{F_w}(x^*, w_{k+1}))|\mathcal{F}_k] \neq 0$, then we would have $\Omega(\lambda) \leq \mathcal{O}(\lambda^2)$, which is a contradiction. It follows that the sequence of random variables given by $[J_{\bar{F}}(x^*) - J_{F_w}(x^*, w_{i+1})](x_i - x^*)\mathbb{1}_{E_i}$ is a Martingale difference and, conditioned on $\mathcal{F}_i$, they are $\hat{\sigma}^2 \|(x_i - x^*)\mathbb{1}_{E_i}\|^2$-subgaussian. Hence, by the definition of the event $E_i$, the random variable $[J_{\bar{F}}(x^*) - J_{F_w}(x^*, w_{i+1})](x_i - x^*)\mathbb{1}_{E_i}$ is $\hat{\sigma}^2 f(\delta, k)\alpha_i$-subgaussian. Thus, Lemma C.1 implies that $\mathbb{1}_{E_{k+1}} \sum_{i=k_0}^{k} [J_{\bar{F}}(x^*) - J_{F_w}(x^*, w_{i+1})](x_i - x^*)$ is $\hat{\sigma}^2 f(\delta, k) \sum_{i=k_0}^{k} \alpha_i$-subgaussian. Furthermore, by Lemma C.2, we have

$$T_4 = \left\{ \mathbb{E} \left[ \exp \left( \frac{6\lambda_k}{(k+1)^2}(1+M) \left\| \sum_{i=k_0}^{k} [J_{\bar{F}}(x^*) - J_{F_w}(x^*, w_{i+1})](x_i - x^*) \right\|_c^2 . \mathbb{1}_{E_{k+1}} \right) \right] \right\}^{1/4}$$

$$\leq \left\{ \frac{1}{\left( 1 - \frac{12 u_{c2}^4 \lambda_k (1+M)}{(k+1)^2} \hat{\sigma}^2 f(\delta, k) \sum_{i=k_0}^{k} \alpha_i \right)^{d/2}} \right\}^{1/4}$$

$$= \frac{1}{\left( 1 - \frac{12 u_{c2}^4 \lambda_k (1+M)}{(k+1)^2} \hat{\sigma}^2 f(\delta, k) \sum_{i=k_0}^{k} \alpha_i \right)^{d/8}}$$

$$\leq \exp \left( \frac{3 u_{c2}^4 d\lambda_k (1+M)}{(k+1)^2} \hat{\sigma}^2 f(\delta, k) \left( \sum_{i=k_0}^{k} \alpha_i \right) \right)$$

$$\leq \exp \left( \frac{3 u_{c2}^4 \alpha d\lambda_k (1+M)}{(k+1)^2} \hat{\sigma}^2 f(\delta, k) \frac{(k+h)^{1-\xi}}{1-\xi} \right) \qquad \text{(integral upper bound)}$$

$$\leq \exp \left( \frac{3 h^{1-\xi} u_{c2}^4 \alpha d\lambda_k (1+M) \hat{\sigma}^2 f(\delta, k)}{(k+1)^{1+\xi}(1-\xi)} \right),$$

where in the second to last inequality we use the fact that $1/(1-x) \leq \exp(2x)$ for all $x \leq 1/2$, and it holds when

$$\frac{12 u_{c2}^4 \alpha \lambda_k (1+M)}{(k+1)^2} \hat{\sigma}^2 f(\delta, k) \frac{(k+h)^{1-\xi} - (k_0+h-1)^{1-\xi}}{1-\xi} \leq \frac{1}{2},$$

which is satisfied because we have

$$\lambda_k \leq \frac{(1-\xi)(k+1)^2}{24 u_{c2}^4 \alpha (1+M) \hat{\sigma}^2 f(\delta, k)(k+h)^{1-\xi}}.$$

Finally, putting everything together, we get

$$\mathbb{P}(\|\tilde{y}_k\|_c^2 \mathbb{1}_{E_{k+1}} \geq \epsilon) \leq e^{-\lambda_k \nu^2 \epsilon} \mathbb{E} \left[ \exp(\lambda_k \nu^2 \|\tilde{y}_k\|_c^2 \mathbb{1}_{E_{k+1}}) \right]$$

$$\leq e^{-\lambda_k \nu^2 \epsilon} \exp\left(\frac{3d\lambda_k u_{c2}^4 \bar{\sigma}^2}{k+1}\right)$$

$$. \exp\left(\frac{9h^\xi(1+M)\lambda_k f(\delta,k)}{2\alpha(k+1)^{2-\xi}}\left(3 + \frac{\xi^2}{(1-\xi/2)^2}\right)\right)$$

$$. \exp\left(\frac{3h^{2-2\xi}\alpha^2\lambda_k N^2(f(\delta,k))^2(1+M)}{2(k+1)^{2\xi}(1-\xi)^2}\right)$$

$$. \exp\left(\frac{3h^{1-\xi}u_{c2}^4\alpha d\lambda_k(1+M)\hat{\sigma}^2 f(\delta,k)}{(k+1)^{1+\xi}(1-\xi)}\right).$$

Making the right hand side equal to $\delta$, and solving for $\epsilon$ we get that, with probability at least $1-\delta$,

$$\|\tilde{y}_k\|^2 \mathbb{1}_{E_{k+1}} \leq \frac{1}{\lambda_k\nu^2}\log(1/\delta) + \frac{3u_{c2}^4 d\bar{\sigma}^2}{\nu^2(k+1)}$$

$$+ \frac{9h^\xi(1+M)f(\delta,k)}{2\alpha(k+1)^{2-\xi}}\left(3 + \frac{\xi^2}{(1-\xi/2)^2}\right)$$

$$+ \frac{3h^{2-2\xi}\alpha^2 N^2(f(\delta,k))^2(1+M)}{2\nu^2(k+1)^{2\xi}(1-\xi)^2}$$

$$+ \frac{3h^{1-\xi}u_{c2}^4\alpha d(1+M)\hat{\sigma}^2 f(\delta,k)}{(k+1)^{1+\xi}(1-\xi)} =: \tilde{\epsilon}(k,\delta). \tag{B.6}$$

Note that, for any two events $A$ and $B$, we have

$$\mathbb{P}(A \cap B) = 1 - \mathbb{P}(A^c \cup B^c) \geq 1 - \mathbb{P}(A^c) - \mathbb{P}(B^c) = \mathbb{P}(A) + \mathbb{P}(B) - 1.$$

Define

$$\epsilon(k,\delta) := \left(\sqrt{\bar{\epsilon}(k,\delta)} + \sqrt{\tilde{\epsilon}(k,\delta)}\right)^2.$$

Then,

$$\mathbb{P}(\|y_k - x^*\|_c^2 \leq \epsilon(k,\delta)) \geq \mathbb{P}\left([\|\bar{y}_k\|_c + \|\tilde{y}_k\|_c]^2 \leq \epsilon(k,\delta)\right) \qquad \text{(Eq. (B.1))}$$

$$\geq \mathbb{P}\left(\{\|\bar{y}_k\|_c^2 \leq \bar{\epsilon}(k,\delta)\} \cap \{\|\tilde{y}_k\|_c^2 \leq \tilde{\epsilon}(k,\delta)\}\right)$$

$$\geq \mathbb{P}(\{\|\tilde{y}_k\|_c^2 \leq \tilde{\epsilon}(k,\delta)\} \cap E_{k+1})$$
$$\qquad (E_{k+1} \subset \{\|\bar{y}_k\|_c \leq \bar{\epsilon}(k,\delta)\} \text{ due to Eq. (B.2)})$$

$$= \mathbb{P}(\{\|\tilde{y}_k\|_c^2 \mathbb{1}_{E_{k+1}} \leq \tilde{\epsilon}(k,\delta)\} \cap E_{k+1})$$
$$\qquad (\text{for all } \omega \in E_{k+1}, \mathbb{1}_{E_{k+1}}(\omega) = 1)$$

$$\geq \mathbb{P}\left(\|\tilde{y}_k\|_c^2 \mathbb{1}_{E_{k+1}} \leq \tilde{\epsilon}(k,\delta)\right) + \mathbb{P}(E_{k+1}) - 1 \qquad \text{(union bound)}$$

$$\geq 1 - \delta - \delta. \qquad \text{(Eq. (B.6) and definition of } E_{k+1})$$

Hence, with probability at least $1-\delta$ (with abuse of notation for substituting $2\delta$ by $\delta$), we have

$$\|y_k - x^*\|_c^2 \leq \left(\sqrt{\tilde{\epsilon}(k,\delta/2)} + \sqrt{\bar{\epsilon}(k,\delta/2)}\right)^2$$

$$= \tilde{\epsilon}(k,\delta/2) + \bar{\epsilon}(k,\delta/2) + 2\sqrt{\bar{\epsilon}(k,\delta/2)\tilde{\epsilon}(k,\delta/2)}$$

$$\leq \frac{g(\delta,k)}{\nu^2(k+1)}\log(2/\delta) + \frac{3u_{c2}^4 d\bar{\sigma}^2}{\nu^2(k+1)}$$

$$+ \frac{9h^\xi(1+M)f(\delta/2,k)}{2\alpha(k+1)^{2-\xi}}\left(3 + \frac{\xi^2}{(1-\xi/2)^2}\right)$$

$$+ \frac{3h^{2-2\xi}\alpha^2 N^2(f(\delta/2,k))^2(1+M)}{2\nu^2(k+1)^{2\xi}(1-\xi)^2}$$

$$+ \frac{3h^{1-\xi}u_{c2}^4\alpha d(1+M)\hat{\sigma}^2 f(\delta/2,k)}{(k+1)^{1+\xi}(1-\xi)}$$

$$+ \frac{\alpha f(\delta,k_0(\delta/2,k)-1)\left[(k_0(\delta/2,k)-1+h)^{1-\xi/2} - (h-1)^{1-\xi/2}\right]^2}{(1-\xi/2)^2(k+1)^2}$$

$$+ \frac{2\sqrt{\alpha f(\delta, k_0(\delta/2, k) - 1)} \left[(k_0(\delta/2, k) - 1 + h)^{1-\xi/2} - (h-1)^{1-\xi/2}\right]}{(1 - \xi/2)(k+1)}$$

$$\times \left[ \frac{g(\delta, k)}{\nu^2(k+1)} \log(2/\delta) + \frac{3u_{c2}^4 d\bar{\sigma}^2}{\nu^2(k+1)} \right.$$

$$+ \frac{9h^\xi(1+M)f(\delta/2, k)}{2\alpha(k+1)^{2-\xi}} \left(3 + \frac{\xi^2}{(1-\xi/2)^2}\right)$$

$$+ \frac{3h^{2-2\xi}\alpha^2 N^2(f(\delta/2, k))^2(1+M)}{2\nu^2(k+1)^{2\xi}(1-\xi)^2}$$

$$\left. + \frac{3h^{1-\xi}u_{c2}^4\alpha d(1+M)\hat{\sigma}^2 f(\delta/2, k)}{(k+1)^{1+\xi}(1-\xi)} \right]^{1/2},$$

where the last inequality follows from (B.2) and (B.6), and

$$g(\delta, k) = 24u_{c2}^4 \max\left\{\bar{\sigma}^2, \frac{\alpha(1+M)\hat{\sigma}^2 h^{1-\xi}f(\delta/2, k)}{(1-\xi)(k+1)^\xi}\right\}.$$

## B.2 Proof of Proposition 4.1

Consider the recursion

$$x_{k+1} = (1 - \alpha_k)x_k + \alpha_k w_{k+1}x_k,$$

where $w_k \sim \mathcal{N}(0, 1)$. It follows that

$$x_k = (1 - \alpha_{k-1} + \alpha_{k-1}w_k)x_{k-1}$$

$$= x_0 \prod_{i=0}^{k-1}(1 - \alpha_i + \alpha_i w_{i+1}).$$

Hence,

$$y_k = \frac{x_0}{k+1}\sum_{k'=0}^{k}\prod_{i=0}^{k'-1}(1 - \alpha_i + \alpha_i w_{i+1}).$$

Without loss of generality, we assume that $x_0 > 0$. Consider the event $E_k = \{w_i \geq 0, i = 1, \ldots, k\}$, and suppose that $k > 2$. For any $t > 0$, we have

$$\exp(ty_k) \geq \mathbb{1}_{E_k}\exp(ty_k)$$

$$\geq \mathbb{1}_{E_k}\exp\left(\frac{tx_2}{k+1}\right)$$

$$= \mathbb{1}_{E_k}\exp\left(x_0 t\frac{(1 - \alpha_0 + \alpha_0 w_1)(1 - \alpha_1 + \alpha_1 w_2)}{k+1}\right)$$

$$\geq \mathbb{1}_{E_k}\exp\left(\frac{x_0\alpha_0\alpha_1 t w_1 w_2}{k+1}\right).$$

Then,

$$\mathbb{E}[\exp(ty_k)] \geq \mathbb{E}\left[\mathbb{1}_{E_k}\exp\left(\frac{x_0\alpha_0\alpha_1 t w_1 w_2}{k+1}\right)\right]$$

$$= \mathbb{E}\left[\mathbb{1}_{E_2}\exp\left(\frac{x_0\alpha_0\alpha_1 t w_1 w_2}{k+1}\right)\right].\mathbb{P}(w_i \geq 0, i = 3, 4, \ldots, k).$$

We also have

$$\mathbb{E}\left[\exp\left(\frac{x_0\alpha_0\alpha_1 t w_1 w_2}{k+1}\right)\right] = \mathbb{E}\left[\mathbb{1}_{E_2}\exp\left(\frac{x_0\alpha_0\alpha_1 t w_1 w_2}{k+1}\right)\right] + \mathbb{E}\left[\mathbb{1}_{\{w_1<0, w_2<0\}}\exp\left(\frac{x_0\alpha_0\alpha_1 t w_1 w_2}{k+1}\right)\right]$$

$$+ \mathbb{E}\left[\mathbb{1}_{\{w_1.w_2<0\}} \exp\left(\frac{x_0\alpha_0\alpha_1 tw_1w_2}{k+1}\right)\right]$$

$$= 2\mathbb{E}\left[\mathbb{1}_{E_2} \exp\left(\frac{x_0\alpha_0\alpha_1 tw_1w_2}{k+1}\right)\right] + \mathbb{E}\left[\mathbb{1}_{\{w_1.w_2<0\}} \exp\left(\frac{x_0\alpha_0\alpha_1 tw_1w_2}{k+1}\right)\right]$$

$$\leq 2\mathbb{E}\left[\mathbb{1}_{E_2} \exp\left(\frac{x_0\alpha_0\alpha_1 tw_1w_2}{k+1}\right)\right] + 1.$$

Hence,

$$\mathbb{E}[\exp(ty_k)] \geq \frac{1}{2}\mathbb{E}\left[\exp\left(\frac{x_0\alpha_0\alpha_1 tw_1w_2}{k+1}\right)\right]\mathbb{P}(w_i \geq 0, i = 3, 4, \ldots, k) - \frac{1}{2}\mathbb{P}(w_i \geq 0, i = 3, 4, \ldots, k)$$

$$= \mathbb{E}\left[\exp\left(\frac{x_0\alpha_0\alpha_1 tw_1w_2}{k+1}\right)\right]\left(\frac{1}{2}\right)^{k-1} - \left(\frac{1}{2}\right)^{k-1},$$

which is infinity for $t > (k+1)/(x_0\alpha_0\alpha_1)$.

Next, consider $\alpha_j = \alpha/(j+h)^\xi$ such that $\alpha_0 < 1/2$. By Markov's inequality, we have

$$\mathbb{P}(|x_i| > \epsilon) \leq \frac{\mathbb{E}[x_i^2]}{\epsilon^2}$$

$$= \frac{x_0^2}{\epsilon^2} \prod_{j=0}^{i-1} \mathbb{E}[(1 - \alpha_j + \alpha_j w_{j+1})^2]$$

$$= \frac{x_0^2}{\epsilon^2} \prod_{j=0}^{i-1} [(1 - \alpha_j)^2 + \alpha_j^2]$$

$$\leq \frac{x_0^2}{\epsilon^2} \exp\left(\sum_{j=0}^{i-1} -2\alpha_j + 2\alpha_j^2\right)$$

$$\leq \frac{x_0^2}{\epsilon^2} \exp\left(-\sum_{j=0}^{i-1} \alpha_j\right)$$

$$\leq \frac{x_0^2}{\epsilon^2} \exp\left(-\frac{\alpha}{(1-\xi)(i-1+h)^{\xi-1}} + \frac{\alpha}{(1-\xi)(h-1)^{\xi-1}}\right)$$

$$\leq \frac{c}{\epsilon^2} \exp\left(-\frac{\alpha}{(1-\xi)(i-1+h)^{\xi-1}}\right),$$

where the last inequality holds for some constant $c > 0$. Then, for $h > 2$, we have

$$\mathbb{P}(|x_i| \leq \epsilon \text{ for all } 0 \leq i \leq k) = 1 - \mathbb{P}(|x_i| > \epsilon \text{ for some } 0 \leq i \leq k)$$

$$\geq 1 - \sum_{i=0}^{k} \mathbb{P}(|x_i| > \epsilon)$$

$$\geq 1 - \frac{c}{\epsilon^2} \sum_{i=0}^{k} \exp\left(-\frac{\alpha}{(1-\xi)(i-1+h)^{\xi-1}}\right)$$

$$\geq 1 - \frac{c}{\epsilon^2} \int_{-1}^{k} \exp\left(-\frac{\alpha}{(1-\xi)(x-1+h)^{\xi-1}}\right) dx$$

$$\geq 1 - \frac{c'}{\epsilon^2} \int_{-1}^{k} \frac{1}{(x-1+h)^{\xi+1}} dx$$

$$\geq 1 - \frac{c''\alpha_k}{\epsilon^2}.$$

Hence, with probability at least $1 - \delta$, for all $0 \le i \le k$, we have $|x_i|^2 \le c'' \alpha_k / \delta$. This means that $f(\delta, k) = c'' / \delta$. Therefore, by Theorem 4.1, with probability at least $1 - \delta$, we have

$$|y_k|^2 \le c \frac{\log(1/\delta) + 1}{k} + o(1/k).$$

The result follows.

## B.3 Proof of Lemma 4.1

The proof follows from Equation (B.2), and by assuming $k_0 = k + 1$. In particular, with probability at least $1 - \delta$, we have

$$
\begin{aligned}
\|y_k - x^*\|_c^2 &\le \frac{\alpha f(\delta, k+1) \left[(k+h)^{1-\xi/2} - (h-1)^{1-\xi/2}\right]^2}{(1 - \xi/2)^2 (k+1)^2} \\
&\le \frac{\alpha f(\delta, k+1)(k+h)^{2-\xi}}{(1 - \xi/2)^2 (k+1)^2} \\
&\le \frac{\alpha h^{2-\xi} f(\delta, k+1)}{(1 - \xi/2)^2 (k+1)^\xi}.
\end{aligned}
$$

## B.4 Proof of Proposition 4.2

First, note that

$$y_k = \frac{x_0 + w_1 + \cdots + w_k}{k+1}.$$

In addition, this example satisfies Assumption 4.3 as

$$
\begin{aligned}
\mathbb{E}\left[\exp\left(\lambda \langle F(x^*, w_{k+1}) - \bar{F}(x^*), v\rangle\right) \big| \mathcal{F}_k\right] &= \mathbb{E}\left[\exp\left(\lambda \langle w_{k+1}, v\rangle\right)\right] \\
&= \mathbb{E}\left[\exp\left(\lambda \sum_j w_{k+1,j} v_j\right)\right] \\
&= \mathbb{E}\left[\exp\left(\lambda \sum_{j \text{ odd}} w_{k+1,j} v_j + \lambda \sum_{j \text{ even}} w_{k+1,j} v_j\right)\right] \\
&= \mathbb{E}\left[\exp\left(\lambda z_{k+1,1} \sum_{j \text{ odd}} v_j + \lambda z_{k+1,2} \sum_{j \text{ even}} v_j\right)\right] \\
&= \mathbb{E}\left[\exp\left(\lambda z_{k+1,1} \sum_{j \text{ odd}} v_j\right)\right] . \mathbb{E}\left[\exp\left(\lambda z_{k+1,2} \sum_{j \text{ even}} v_j\right)\right] \\
&= \mathbb{E}\left[\exp\left(\frac{\bar{\sigma}^2 \lambda^2}{2d} \left(\sum_{j \text{ odd}} v_j\right)^2\right)\right] . \mathbb{E}\left[\exp\left(\frac{\bar{\sigma}^2 \lambda^2}{2d} \left(\sum_{j \text{ even}} v_j\right)^2\right)\right] \\
&\le \mathbb{E}\left[\exp\left(\frac{\bar{\sigma}^2 \lambda^2 \|v\|_1^2}{2d}\right)\right] \\
&\le \mathbb{E}\left[\exp\left(\frac{\bar{\sigma}^2 \lambda^2 \|v\|_2^2}{2}\right)\right].
\end{aligned}
$$

For the sake of simplicity, we assume $x_0 = 0$. Then, we have

$$
\begin{aligned}
\mathbb{P}\left(\frac{\sqrt{k \bar{\sigma}^2 \log(1/\delta)} - \|x_0\|_2}{k+1} \ge \|y_k - x^*\|_2\right) &\\
&= \mathbb{P}\left(\sqrt{k \bar{\sigma}^2 \log(1/\delta)} - \|x_0\|_2 \ge (k+1)\|y_k\|_2\right)
\end{aligned}
$$

$$= \mathbb{P}\left(\sqrt{k\bar{\sigma}^2 \log(1/\delta)} - \|x_0\|_2 \geq \left\|x_0 + \sum_{i=1}^{k} w_i\right\|_2\right)$$

$$\leq \mathbb{P}\left(\sqrt{k\bar{\sigma}^2 \log(1/\delta)} - \|x_0\|_2 \geq -\|x_0\|_2 + \left\|\sum_{i=1}^{k} w_i\right\|_2\right)$$

(triangle inequality)

$$= \mathbb{P}\left(\sqrt{k\bar{\sigma}^2 \log(1/\delta)} \geq \left\|\sum_{i=1}^{k} w_i\right\|_2\right)$$

$$= \mathbb{P}\left(\sqrt{k\bar{\sigma}^2 \log(1/\delta)} \geq \sqrt{\sum_{j=1}^{d}\left(\sum_{i=1}^{k} w_{i,j}\right)^2}\right)$$

$$= \mathbb{P}\left(\sqrt{k\bar{\sigma}^2 \log(1/\delta)} \geq \sqrt{d\frac{\left(\sum_{i=1}^{k} z_{i,1}\right)^2 + \left(\sum_{i=1}^{k} z_{i,2}\right)^2}{2}}\right)$$

$$= \mathbb{P}\left(\sqrt{2\log(1/\delta)} \geq \sqrt{\frac{d\left\|\sum_{i=1}^{k} z_i\right\|_2^2}{\bar{\sigma}^2 k}}\right)$$

$$= \int_{0}^{2\pi} \int_{0}^{\sqrt{2\log(1/\delta)}} \frac{1}{2\pi} r e^{-r^2/2} dr d\theta = 1 - \delta.$$

## B.5   Proof of Theorem 5.1

Since the $c$-norm can be non-smooth, to prove a concentration result on the error we use the generalized Moreau envelope

$$M(x) = \min_{u\in\mathbb{R}^d}\left\{\frac{1}{2}\|u\|_c^2 + \frac{1}{2\mu}\|x - u\|_s^2\right\},$$

where $\|\cdot\|_s$ is an arbitrary smooth norm. Then, we have that $M(\cdot)$ is an $L/\mu$ – smooth function with respect to $\|\cdot\|_s$. Moreover, the Moreau envelope defines the norm $\|\cdot\|_M = \sqrt{2M(\cdot)}$. For a more detailed discussion about this function, please refer to [12].

Since Assumptions 5.1, and 5.2 are satisfied, and $\alpha_i = \alpha/(i + h)^\xi$, where $\xi \in (0, 1)$. Then, by choosing $\alpha > 0$ and $h \geq (2\xi/[(1 - \tilde{\gamma}_c)\alpha])^{1/(1-\xi)}$, for any $\delta > 0$ and $K \geq 0$, using the result from [14, Theorem 2.4], we know that with probability at least $1 - \delta$, we have for all $i \geq K$ that

$$\|x_i - x^*\|_c^2 \leq \frac{\bar{c}_1 \log(1/\delta)}{(i + h)^\xi} + \bar{c}_2\|x_0 - x^*\|_c^2 \exp\left(-\frac{\bar{D}\alpha((i + h)^{1-\xi} - h^{1-\xi})}{2(1 - \xi)}\right)$$
$$+ \frac{\bar{c}_5 + \bar{c}_6 \log((i + 1)/K^{1/2})}{(i + h)^\xi},$$

(B.7)

where

$$\bar{c}_1 = \frac{16\sigma^2 u_{M,c*}^2 u_{cM}^2 \alpha}{1 - \tilde{\gamma}_c}$$

$$\bar{c}_2 = \frac{u_{cM}^2}{\ell_{cM}^2}$$

$$\bar{c}_3 = \frac{32ec_d\sigma^2 Lu_{cM}^2\alpha^2}{((1-\tilde{\gamma}_c)\alpha-1)\mu\ell_{cs}^2}$$

$$\bar{c}_4 = \frac{16\sigma^2 c_d Lu_{cM}^2\alpha^2}{\mu\ell_{cs}^2}$$

$$\bar{c}_5 = \frac{16eu_{cM}^2 c_d\sigma^2 L\alpha}{\mu\ell_{cs}^2(1-\tilde{\gamma}_c)}$$

$$\bar{c}_6 = \frac{32u_{cM}^2\sigma^2 u_{M,c*}^2\alpha}{1-\tilde{\gamma}_c},$$

with $\tilde{\gamma}_c = \gamma_c(1+\mu u_{cs}^2)^{1/2}/(1+\mu\ell_{cs}^2)^{1/2}$ and $\bar{D} = 2(1-\tilde{\gamma}_c)$. Next, choosing $K = 1$, we get that Equation (B.7) holds for all $i \geq 1$. In addition, since $\bar{c}_2 \geq 1$ and $\bar{c}_l \geq 0$ for $l = 1, 4, 5$ this bound holds $i = 0$ as well. Hence, for all $i \leq k$, we have

$$\|x_i - x^*\|_c^2 \leq \alpha_i\left(\frac{\bar{c}_1\log(1/\delta)}{\alpha} + \frac{\bar{c}_2\|x_0-x^*\|_c^2(i+h)^\xi}{\alpha}\exp\left(-\frac{\bar{D}\alpha((i+h)^{1-\xi}-h^{1-\xi})}{2(1-\xi)}\right) + \frac{\bar{c}_5+\bar{c}_6\log(i+1)}{\alpha}\right)$$

$$\leq \alpha_i\underbrace{\left(\frac{\bar{c}_1\log(1/\delta)}{\alpha} + d_1 + \frac{\bar{c}_5+\bar{c}_6\log(k+1)}{\alpha}\right)}_{f_\xi(\delta,k)}, \tag{B.8}$$

where

$$d_1 := \sup_{i\geq 0}\left\{\frac{\bar{c}_2\|x_0-x^*\|_c^2(i+h)^\xi}{\alpha}\exp\left(-\frac{\bar{D}\alpha((i+h)^{1-\xi}-h^{1-\xi})}{2(1-\xi)}\right)\right\} < \infty.$$

We get the result by directly applying Theorem 4.1.

## B.6   Proof of Theorem 5.1

From the proof of Theorem 3 in [37], for $h$ large enough, we have

$$\mathbb{E}[u_{i+1}|\mathcal{F}_i] \leq (1-\beta_1\alpha_i)u_i + \beta_2\alpha_i^3, \tag{B.9}$$

where $u_i = \|x_{i+1}-x^*\|_c^4 + \beta_3\alpha_i\|x_{i+1}-x^*\|_c^2$ for some positive constants $\beta_1, \beta_2$, and $\beta_3$. Define

$$M_i = \begin{cases} \frac{1}{\alpha_i^2}u_i + 4\beta_2\sum_{j=i}^k \alpha_j & i \leq k \\ M_k & i > k. \end{cases}$$

Next, we prove that $\{M_i\}_{i\geq 0}$ is a supermatringale:

$$\mathbb{E}[M_{i+1}|\mathcal{F}_i] \leq \mathbb{E}\left[\frac{(1-\beta_2\alpha_i)u_i + \beta_4\alpha_i^3}{\alpha_{i+1}^2} + 4\beta_4\sum_{j=i+1}^k \alpha_j\right] \qquad \text{(by Eq. (B.9))}$$

$$\leq \mathbb{E}\left[\frac{(1-\beta_2\alpha_i)u_i}{\alpha_{i+1}^2} + 4\beta_4\sum_{j=i}^k \alpha_j\right] \qquad \text{(by } \alpha_i^2/\alpha_{i+1}^2 \leq 4)$$

$$\leq \mathbb{E}\left[\frac{u_i}{\alpha_i^2} + 4\beta_4\sum_{j=i}^k \alpha_j\right]$$

$$= M_i.$$

In the last inequality, we used the following

$$\frac{\alpha_i^2}{\alpha_{i+1}^2}\left(1-\beta_2\alpha_i\right) = \left(\frac{i+h+1}{i+h}\right)^\xi\left(1-\frac{\beta_2\alpha}{(i+h)^\xi}\right)$$

$$\leq \left(\frac{i+h+1}{i+h}\right)^{2\xi}\exp\left(-\frac{\beta_2\alpha}{(i+h)^\xi}\right)$$

$$= \left[ \left( 1 + \frac{1}{i+h} \right)^{i+h} \right]^{2\xi/(i+h)} \exp\left( -\frac{\beta_2 \alpha}{(i+h)^\xi} \right)$$

$$\leq \exp\left( \frac{2\xi}{i+h} - \frac{\beta_2 \alpha}{(i+h)^\xi} \right)$$

$$\leq 1. \qquad\qquad\qquad \text{(for } h \text{ large enough.)}$$

Hence, by Ville's maximal inequality, we have

$$\mathbb{P}\left( \sup_{i \geq 0} M_i \geq \epsilon \right) \leq \frac{\mathbb{E}[M_0]}{\epsilon}$$

$$= \frac{u_0/\alpha_0^2 + 4\beta_4 \sum_{j=0}^{k} \alpha_j}{\epsilon}$$

Hence,

$$\mathbb{P}\left( \sup_{i \geq 0} M_i < \epsilon \right) > 1 - \underbrace{\frac{u_0/\alpha_0^2 + 4\beta_4 \sum_{j=0}^{k} \alpha_j}{\epsilon}}_{\delta}.$$

By a change of variables, we get

$$\mathbb{P}\left( \sup_{i \geq 0} M_i < \frac{u_0/\alpha_0^2 + 4\beta_4 \sum_{j=0}^{k} \alpha_j}{\delta} \right) > 1 - \delta.$$

Hence,

$$1 - \delta < \mathbb{P}\left( M_i < \frac{u_0/\alpha_0^2 + 4\beta_4 \sum_{j=0}^{k} \alpha_j}{\delta}, \forall i \geq 0 \right)$$

$$< \mathbb{P}\left( \frac{1}{\alpha_i^2} \| x_i - x^* \|_c^4 < \frac{u_0/\alpha_0^2 + 4\beta_4 \sum_{j=0}^{k} \alpha_j}{\delta}, \forall 0 \leq i \leq k \right)$$

$$= \mathbb{P}\left( \frac{1}{\alpha_i} \| x_i - x^* \|_c^2 < \sqrt{\frac{u_0/\alpha_0^2 + 4\beta_4 \sum_{j=0}^{k} \alpha_j}{\delta}}, \forall 0 \leq i \leq k \right)$$

Applying Theorem 4.1 with $f_\xi(\delta, k) = \sqrt{\frac{u_0/\alpha_0^2 + 4\beta_4 \sum_{j=0}^{k} \alpha_j}{\delta}} = \frac{\mathcal{O}(k^{1/2 - \xi/2})}{\delta^{1/2}}$, we get the result.

## B.7   Proof of Theorem 5.2

The idea of this proof is borrowed from the example in [14, Example 2.2].

Consider the SA presented in Equation (1.1) for a 1-dimensional linear setting with $F(x, w) = wx$. In this case, let $\{w_{k+1}\}_{k \geq 0}$ be an i.i.d. sequence of real-valued random variables such that $\mathbb{P}(w_k = a + N) = 1/(N+1)$ and $\mathbb{P}(w_k = a - 1) = N/(N+1)$, where $a \in (0, 1)$ and $N \geq 1$ are tunable parameters. Note that the update equation can be equivalently written as

$$x_{k+1} = (1 + (w_{k+1} - 1)\alpha_k)x_k. \tag{B.10}$$

In this example, we have $x_k \in \mathbb{R}^+$ for all $k \geq 0$ and $x^* = 0$. To begin with, there exists $k_0 > 0$ such that

$$\exp\left( \frac{\alpha_k(a + N - 1)}{2} \right) \leq 1 + \alpha_k(a + N - 1), \quad \forall k \geq k_0. \tag{B.11}$$

As a result, we have for any $\lambda > 0$ and $k$ large enough that

$$\mathbb{E}\left[ \exp\left( \lambda(k+1)^{\tilde{\beta}'} y_k^{\tilde{\beta}} \right) \right]$$

$$= \mathbb{E}\left[\exp\left(\lambda x_0^{\tilde{\beta}}(k+1)^{\tilde{\beta}'}\left[\frac{1}{k+1}\sum_{j=0}^{k}x_j\right]^{\tilde{\beta}}\right)\right]$$

$$= \mathbb{E}\left[\exp\left(\lambda x_0^{\tilde{\beta}}(k+1)^{\tilde{\beta}'}\left[\frac{1}{k+1}\sum_{j=0}^{k}\prod_{i=0}^{j-1}(1+\alpha_i(w_{i+1}-1))\right]^{\tilde{\beta}}\right)\right]$$

$$\overset{(a)}{\geq} \frac{1}{(N+1)^k}\exp\left(\lambda x_0^{\tilde{\beta}'}(k+1)^{\tilde{\beta}'}\left[\frac{1}{k+1}\sum_{j=0}^{k}\prod_{i=0}^{j-1}(1+\alpha_i(a+N-1))\right]^{\tilde{\beta}}\right)$$

$$\overset{(b)}{\geq} \frac{1}{(N+1)^k}\exp\left(\lambda x_0^{\tilde{\beta}'}(k+1)^{\tilde{\beta}'}\left[\frac{1}{k+1}\sum_{j=k_0+1}^{k}\prod_{i=k_0}^{j-1}(1+\alpha_i(a+N-1))\right]^{\tilde{\beta}}\right)$$

$$\overset{(c)}{\geq} \frac{1}{(N+1)^k}\exp\left(\lambda x_0^{\tilde{\beta}}(k+1)^{\tilde{\beta}'}\left[\frac{1}{k+1}\sum_{j=k_0+1}^{k}\exp\left(\frac{1}{2}\sum_{i=k_0}^{j-1}\alpha_i(a+N-1)\right)\right]^{\tilde{\beta}}\right)$$

$$\overset{(d)}{\geq} \exp\left(\lambda x_0^{\tilde{\beta}}(k+1)^{\tilde{\beta}'}\left[\frac{1}{k+1}\sum_{j=k_0+1}^{k}\exp\left(\frac{\alpha(a+N-1)}{2(1-\xi)}((j+h)^{1-\xi}-(k_0+h)^{1-\xi})\right)\right]^{\tilde{\beta}}-k\ln(N+1)\right)$$

$$\overset{(e)}{\geq} \exp\left(\lambda x_0^{\tilde{\beta}}(k+1)^{\tilde{\beta}'}\left[\frac{1}{(k+1)\left\lceil 2/(\tilde{\beta}(1-\xi))\right\rceil!}\right.\right.$$
$$\left.\left.\times\left[\sum_{j=k_0+1}^{k}\left(\frac{\alpha(a+N-1)}{2(1-\xi)}((j+h)^{1-\xi}-(k_0+h)^{1-\xi})\right)^{\left\lceil 2/(\tilde{\beta}(1-\xi))\right\rceil}\right]\right]^{\tilde{\beta}}-k\ln(N+1)\right)$$

$$\overset{(f)}{\geq} \exp\left(\lambda x_0^{\tilde{\beta}}(k+1)^{\tilde{\beta}'}\left[\frac{1}{(k+1)\left\lceil 2/(\tilde{\beta}(1-\xi))\right\rceil!}\right.\right.$$
$$\left.\left.\times\left[\sum_{j=k_1}^{k}\left(\frac{\alpha(a+N-1)}{2(1-\xi)}((j+h)^{1-\xi}-(k_0+h)^{1-\xi})\right)^{2/(\tilde{\beta}(1-\xi))}\right]\right]^{\tilde{\beta}}-k\ln(N+1)\right)$$

$$\overset{(g)}{\geq} \exp\left(\lambda x_0^{\tilde{\beta}}(k+1)^{\tilde{\beta}'}\left[\frac{1}{(k+1)\left\lceil 2/(\tilde{\beta}(1-\xi))\right\rceil!}\right.\right.$$
$$\left.\left.\times\left[\int_{k_1}^{k}\left(\frac{\alpha(a+N-1)}{2(1-\xi)}((u+h)^{1-\xi}-(k_0+h)^{1-\xi})\right)^{2/(\tilde{\beta}(1-\xi))}du\right]\right]^{\tilde{\beta}}-k\ln(N+1)\right)$$

$$\overset{(h)}{\geq} \exp\left(\lambda x_0^{\tilde{\beta}}(k+1)^{\tilde{\beta}'}\left[\frac{1}{(k+1)\left\lceil 2/(\tilde{\beta}(1-\xi))\right\rceil!}\int_{k_1}^{k}\left(\frac{\alpha(a+N-1)}{2(1-\xi)}((u+h)^{2/\tilde{\beta}}\right.\right.\right.$$
$$\left.\left.\left.-\left(2/(\tilde{\beta}(1-\xi))\right)(u+h)^{2/\tilde{\beta}-1+\xi}(k_0+h)^{1-\xi}\right)du\right]^{\tilde{\beta}}-k\ln(N+1)\right)$$

$$\geq \exp\left(\theta(k^{\tilde{\beta}'}[k^{2/\tilde{\beta}}]^{\tilde{\beta}}) - k\ln(N+1)\right) = \exp\left(\theta(k^{2+\tilde{\beta}'})\right),$$

where $(a)$ is by lower bounding expectation with one of its terms, $(b)$ is by $a + N - 1 > 0$, $(c)$ is by Equation (B.11), $(d)$ follows from

$$\sum_{i=k_0}^{k-1} \alpha_i \geq \int_{k_0}^{k} \frac{\alpha}{(x+h)^\xi} dx = \frac{\alpha}{1-\xi}((k+h)^{1-\xi} - (k_0+h)^{1-\xi}), \tag{B.12}$$

$(e)$ is due to the fact that for any $r > 0$, $e^r \geq r^m/m!$ for all $m \in \mathbb{N}$, $(f)$ is due to picking $k_1 \geq k_0 + 1$ such that

$$\frac{\alpha(a+N-1)}{2(1-\xi)}((k_1+h)^{1-\xi} - (k_0+h)^{1-\xi}) > 1,$$

$(g)$ is due to an integral lower bound similar to Equation (B.12), and $(h)$ is due to Bernoulli's inequality. Therefore, for any $\tilde{\beta} > 0$ and $\tilde{\beta}' > 0$, we have

$$\liminf_{k\to\infty} \mathbb{E}\left[\exp\left(\lambda(k+1)^{\tilde{\beta}'} y_k^{\tilde{\beta}}\right)\right] = \infty, \quad \forall \lambda > 0.$$

As a result, according to Lemma [14, Lemma 4.1], there do not exist $\bar{K}'_1, \bar{K}'_2 > 0$ such that

$$\mathbb{P}\left((k+1)^{\tilde{\beta}'/\tilde{\beta}} y_k \geq \epsilon\right) \leq \bar{K}'_1 \exp\left(-\bar{K}'_2 \epsilon^{\tilde{\beta}}\right), \quad \forall \epsilon > 0, k \geq 0.$$

A change of variables finishes the proof.

## B.8   Proof of Theorem 6.1

First, we verify that the assumptions of Theorem 5.1 are satisfied.

- **Assumption 5.1:** Firstly, for tabular TD$(n)$ algorithm we assumed that $V_k(s) \leq R_{\max}/(1-\gamma)$ for all $s \in \mathcal{S}$ and $k \geq 0$. Furthermore, TD$(n)$ can be written in the form of (1.1) with

$$(F(V_k, w_{k+1}))_s = \mathbb{1}_{\{s=S_k^0\}}\left(\sum_{i=0}^{n-1}\gamma^i(\mathcal{R}(S_k^i, A_k^i) + \gamma V_k(S_k^{i+1}) - V_k(S_k^i))\right) + V_k(s)$$

$$= \mathbb{1}_{\{s=S_k^0\}}\left(\gamma^n V_k(S_k^n) - V_k(s) + \sum_{i=0}^{n-1}\gamma^i\mathcal{R}(S_k^i, A_k^i)\right) + V_k(s), \tag{B.13}$$

where $w_{k+1} = [(S_k^i, A_k^i)]_{0\leq i\leq n}$. In addition, we have

$$(\bar{F}(V_k))_s = \mu^\pi(s)\sum_{s'}\left(\gamma^n(P^\pi)_{s,s'}^n V_k(s') + \sum_{i=0}^{n-1}\gamma^i(P^\pi)_{s,s'}^i\sum_a \mathcal{R}(s', a)\pi(a\,|\,s)\right) + (1-\mu^\pi(s))V_k(s),$$

where the matrix $P^\pi$ is the transition probability of the Markov chain following policy $\pi$, i.e., we have

$$P_{s,s'}^\pi = \sum_a P(s'\,|\,s, a)\pi(a\,|\,s).$$

Furthermore, we have

$$\bar{F}(V) = M^\pi(\mathcal{T}^\pi)^n(V) + (I - M^\pi)V,$$

where $M^\pi$ is a diagonal matrix with diagonal entries $\mu^\pi$, and $\mathcal{T}^\pi$ is the Bellman operator. Hence, we can write

$$\bar{F}(V) = \underbrace{(I - M^\pi(I - \gamma^n(P^\pi)^n))}_{A^\pi}V + b^\pi.$$

Moreover, we have

$$\sum_{s'} A_{s,s'}^\pi = \sum_{s'}[\delta_{s,s'} - (M^\pi(I - \gamma^n(P^\pi)^n))_{s,s'}]$$

$$= \sum_{s'} \left[ \delta_{s,s'} - \sum_{s''} [M^\pi_{ss''}(I - \gamma^n (P^\pi)^n)_{s''s'}] \right]$$

$$= \sum_{s'} \left[ \delta_{s,s'} - \sum_{s''} [\delta_{ss''} \mu^\pi(s)(\delta_{s''s'} - \gamma^n (P^\pi)^n_{s''s'})] \right]$$

$$= \sum_{s'} [\delta_{s,s'} - [\mu^\pi(s)(\delta_{ss'} - \gamma^n (P^\pi)^n_{ss'})]]$$

$$= 1 - \mu^\pi(s) + \gamma^n \sum_{s'} \mu^\pi(s)(P^\pi)^n_{ss'}$$

$$= 1 - \mu^\pi(s) + \gamma^n \mu^\pi(s)$$

$$= 1 - (1 - \gamma^n)\mu^\pi(s). \tag{B.14}$$

Now, consider the matrix $B^\pi \in \mathbb{R}^{|\mathcal{S}| \times |\mathcal{S}|}$, where $B^\pi_{ss'} = (1 - \gamma^n)\mu^\pi(s)/|\mathcal{S}|$. Note that $C^\pi = A^\pi + B^\pi$ is a stochastic matrix, and its corresponding stationary distribution is $\nu^\pi$ such that $(\nu^\pi)^\top C^\pi = (\nu^\pi)^\top$. Further, we have

$$\nu^\pi_s = \sum_{s'} \nu^\pi_{s'} C^\pi_{ss'}$$

$$= \sum_{s'} \nu^\pi_{s'} (A^\pi_{ss'} + B^\pi_{ss'})$$

$$\geq \sum_{s'} \nu^\pi_{s'} \frac{(1 - \gamma^n)\mu^\pi(s)}{|\mathcal{S}|}$$

$$\geq \frac{(1 - \gamma^n)\mu^\pi_{\min}}{|\mathcal{S}|}, \tag{B.15}$$

where $\mu_{\min} = \min_s\{\mu^\pi(s)\}$. Let us pick an arbitrary $p \geq 2$, and denote $\|\cdot\|_{\nu^\pi, p}$ as the weighted $p$-norm with weights $\nu^\pi$. We then have

$$\|\bar{F}(V_1) - \bar{F}(V_2)\|^p_{\nu^\pi, p} = \|A^\pi(V_1 - V_2)\|^p_{\nu^\pi, p}$$

$$= \sum_s \nu^\pi(s)(A^\pi(V_1 - V_2))^p_s$$

$$= \sum_s \nu^\pi(s) \left( \sum_{s'} A^\pi_{ss'}(V_1 - V_2)(s') \right)^p$$

$$= \sum_s \nu^\pi(s) \left[ \sum_{s''} A^\pi_{ss''} \right]^p \left( \sum_{s'} \frac{A^\pi_{ss'}}{\sum_{s''} A^\pi_{ss''}} (V_1 - V_2)(s') \right)^p$$

$$\leq \sum_s \nu^\pi(s) \left[ \sum_{s''} A^\pi_{ss''} \right]^p \sum_{s'} \frac{A^\pi_{ss'}}{\sum_{s''} A^\pi_{ss''}} ((V_1 - V_2)(s'))^p$$

(Jensen's inequality)

$$= \sum_s \nu^\pi(s) \left[ \sum_{s''} A^\pi_{ss''} \right]^{p-1} \sum_{s'} A^\pi_{ss'} ((V_1 - V_2)(s'))^p$$

$$= \sum_s \nu^\pi(s) [1 - (1 - \gamma^n)\mu^\pi(s)]^{p-1} \sum_{s'} A^\pi_{ss'} ((V_1 - V_2)(s'))^p$$

(by (B.14))

$$\leq \sum_s \nu^\pi(s) [1 - (1 - \gamma^n)\mu^\pi(s)]^{p-1} \sum_{s'} C^\pi_{ss'} ((V_1 - V_2)(s'))^p$$

$$\leq [1 - (1 - \gamma^n)\mu^\pi_{\min}]^{p-1} \sum_s \nu^\pi(s) \sum_{s'} C^\pi_{ss'} ((V_1 - V_2)(s'))^p$$

$$= [1 - (1 - \gamma^n)\mu^\pi_{\min}]^{p-1} \sum_{s'} \nu^\pi(s') ((V_1 - V_2)(s'))^p$$

($\nu^\pi$ is stationary distribution)

$$= [1 - (1 - \gamma^n)\mu_{\min}^\pi]^{p-1} \|V_1 - V_2\|_{\nu^\pi,p}^p.$$

Therefore, $\bar{F}(V)$ is a contraction with respect to $\|\cdot\|_{\nu^\pi,p}$ with contraction factor $\gamma_c = [1 - (1 - \gamma^n)\mu_{\min}^\pi]^{1-1/p}$, and hence Assumption 5.1 is satisfied.

- **Assumption 5.2:** Since the state and action space are both bounded, the noise in the update of TD-learning is also bounded. Hence, Equation (5.1) is satisfied for some $\sigma > 0$ by Hoeffding's lemma. In addition, Equation (5.2) is satisfied due to Lemma C.2.

- **Assumption 4.1:** This assumption is satisfied with $M = p - 1$ due to Lemma C.4.

- **Assumption 4.2:** Since $\bar{F}$ is a contraction, Banach's fixed point theorem implies it has a unique fixed point, and Lemma C.3 implies that $J_{\bar{F}}(x^*) - I$ is invertible. Moreover, since $F$ is linear, it is $(0, \infty)$-locally psuedo smooth.

- **Assumption 4.3:** We have

$$|(F(V^\pi, w_{k+1}) - \bar{F}(V^\pi))_s| \leq \frac{R_{\max}}{1-\gamma} \mathbb{1}_{\{s=s_k^0\}}.$$

Hence,

$$|\langle F(V^\pi, w_{k+1}) - \bar{F}(V^\pi), v \rangle| \leq \|F(V^\pi, w_{k+1}) - \bar{F}(V^\pi)\|_{\nu^\pi,p}^* \cdot \|v\|_{\nu^\pi,p}$$

$$\text{(Hölder's inequality)}$$

$$\leq (\nu_{\min}^\pi)^{-1/p} \frac{R_{\max}}{1-\gamma} \|v\|_{\nu^\pi,p},$$

where in the last inequality we used the fact the

$$\|x\|_{\nu^\pi,p}^* = \left( \sum_i (\nu^\pi(i))^{-\frac{q}{p}} |x_i|^q \right)^{1/q},$$

where $q$ is such that $1/p + 1/q = 1$, and $\nu_{\min}^\pi = \min_s\{\nu^\pi(s)\}$. It follows that

$$\mathbb{E}\left[ \exp\left( \lambda \langle F(V^\pi, w_{k+1}) - \bar{F}(V^\pi), v \rangle \right) \big| \mathcal{F}_k \right] \leq \exp\left( \lambda^2 \frac{R_{\max}^2 \|v\|_{\nu^\pi,p}^2}{2(1-\gamma)^2 \nu_{\min}^{2/p}} \right)$$

$$\text{(Hoffding's lemma)}$$

$$\leq \exp\left( \lambda^2 \frac{R_{\max}^2 \|v\|_{\nu^\pi,p}^2 |\mathcal{S}|^{2/p}}{2(1-\gamma)^2 (\mu_{\min}^\pi (1 - \gamma^n))^{2/p}} \right).$$

$$\text{(Eq. (B.15))}$$

Therefore, the first part of Assumption 4.3 is satisfied with

$$\bar{\sigma}^2 = \frac{R_{\max}^2 |\mathcal{S}|^{2/p}}{(1-\gamma)^2 (\mu_{\min}^\pi (1 - \gamma^n))^{2/p}}.$$

On the other hand, by the definition in (B.13), we have $F(V, w_{k+1}) = A(w_{k+1})V + b(w_{k+1})$. Hence, $J_{F_w}(V^\pi, w_{k+1}) = A(w_{k+1})$. Since we assume finite state and action spaces, and $w_{k+1}$ can only take finitely many values, therefore, by Hoeffding's lemma there exist $\hat{\sigma} < \infty$ such that the second part of Assumption 4.3 holds.

Since $\|x\|_{\nu^\pi,p} \leq \|x\|_p \leq \|x\|_2$, we have $u_{c2} = 1$. By a direct application of Theorem 5.1, we have with probability at least $1 - \delta$,

$$\|\bar{V}_k - V^\pi\|_{\nu^\pi,p}^2 \leq \frac{R_{\max}^2 |\mathcal{S}|^{2/p}}{(1-\gamma)^2 (\mu_{\min}^\pi (1 - \gamma^n))^{2/p}} \frac{24 \log(2/\delta) + 3d}{(1-\gamma_c)^2 (k+1)} + \mathcal{O}(p)\tilde{\mathcal{O}}\left( \frac{1}{k^{3/2}} \right) \mathcal{O}(\log(1/\delta)).$$

Moreover, we have

$$\|y_k - x^*\|_{\nu^\pi,p}^2 \geq (\nu_{\min}^\pi)^{2/p} \|y_k - x^*\|_\infty^2$$

$$\geq \left( \frac{(1 - \gamma^n)\mu_{\min}^\pi}{|\mathcal{S}|} \right)^{2/p} \|y_k - x^*\|_\infty^2$$

Hence, with probability $1 - \delta$, we have

$$\|\bar{V}_k - V^\pi\|_\infty^2 \leq \frac{R_{\max}^2}{(1-\gamma)^2} \left( \frac{|\mathcal{S}|}{\mu_{\min}^\pi (1 - \gamma^n)} \right)^{4/p} \frac{24 \log(2/\delta) + 3d}{(1 - \gamma_c)^2 (k+1)} + \mathcal{O}(p) \tilde{\mathcal{O}} \left( \frac{1}{k^{3/2}} \right) \mathcal{O}(\log(1/\delta)). \tag{B.16}$$

This result holds for any $p$. In particular, by choosing

$$p = \frac{-4 \log \left( 1 - (1 - \gamma^n) \mu_{\min}^\pi \right)}{(1 - \gamma^n) \mu_{\min}^\pi} \log \left( \frac{|\mathcal{S}|}{\mu_{\min}^\pi (1 - \gamma^n)} \right) (k+1)^{1/4},$$

we have

$$\left( \frac{|\mathcal{S}|}{\mu_{\min}^\pi (1 - \gamma^n)} \right)^{4/p} = \exp \left( \frac{4}{p} \log \left( \frac{|\mathcal{S}|}{\mu_{\min}^\pi (1 - \gamma^n)} \right) \right)$$

$$\leq \exp \left( \frac{1}{(k+1)^{1/4}} \right) \quad (-\log(1 - (1 - \gamma^n) \mu_{\min}^\pi) > (1 - \gamma^n) \mu_{\min}^\pi)$$

$$\leq 1 + \frac{2}{(k+1)^{1/4}}. \tag{B.17}$$

In addition, we have

$$\frac{1}{1 - \gamma_c} = \frac{1}{1 - [1 - (1 - \gamma^n) \mu_{\min}^\pi]^{1 - \frac{1}{p}}}$$

$$\leq \frac{1}{(1 - \gamma^n) \mu_{\min}^\pi} + \frac{1 - (1 - \gamma^n) \mu_{\min}^\pi}{2(1 - \gamma^n) \mu_{\min}^\pi \log \left( \frac{|\mathcal{S}|}{\mu_{\min}^\pi (1 - \gamma^n)} \right) (k+1)^{1/4}}, \tag{B.18}$$

where in the last inequality we used the fact that for all $0 < a < 1$, we have

$$\frac{1}{1 - a^{1-x}} \leq \frac{1}{1 - a} + 2 \frac{a \log(1/a)}{(1 - a)^2} x$$

for all $x \leq (1 - a)/(2 \log(1/a))$, and we take $x = 1/p$ and $a = 1 - (1 - \gamma^n) \mu_{\min}^\pi$. Substituting equations (B.17) and (B.18) in Equation (B.16), we obtain

$$\|\bar{V}_k - V^\pi\|_\infty^2 \leq \frac{R_{\max}^2}{(1-\gamma)^2} \left( 1 + \frac{2}{(k+1)^{1/4}} \right) \frac{24 \log(2/\delta) + 3d}{(k+1)} \left( \frac{1}{(1 - \gamma^n) \mu_{\min}^\pi} + \mathcal{O} \left( \frac{1}{k^{1/4}} \right) \right)^2$$

$$\qquad + \tilde{\mathcal{O}} \left( \frac{1}{k^{5/4}} \right) \mathcal{O}(\log(1/\delta))$$

$$= \frac{R_{\max}^2}{(1-\gamma)^2 (1 - \gamma^n)^2 (\mu_{\min}^\pi)^2} \frac{24 \log(2/\delta) + 3d}{(k+1)} + \tilde{\mathcal{O}} \left( \frac{1}{k^{5/4}} \right) \mathcal{O}(\log(1/\delta))$$

## B.9  Proof of Theorem 6.2

First, we verify that the assumptions of Theorem 5.1 are satisfied.

- **Assumption 5.1:** Firstly, for $Q$-learning algorithm it is known that $Q_k(s, a) \leq R_{\max}/(1-\gamma)$ for all $s \in \mathcal{S}$ and $k \geq 0$. Furthermore, $Q$-learning can be written in the form of Equation (1.1) with

  $$(F(Q_k, w_{k+1}))_{s,a} = \mathbb{1}_{\{s = S_k, a = A_k\}} \left( \mathcal{R}(S_k, A_k) + \gamma \max_{a'} \{Q_k(S_k', a')\} - Q_k(S_k, A_k) \right) + Q_k(s, a),$$

  where $w_{k+1} = (S_k, A_k, S_k')$. In addition, we have

  $$(\bar{F}(Q_k))_{s,a} = \mu^{\pi_b}(s) \pi_b(a \,|\, s) \left( \mathcal{R}(s, a) + \gamma \sum_{s'} P(s'|s, a) \max_{a'} \{Q_k(s', a')\} \right) + (1 - \mu^{\pi_b}(s) \pi_b(a \,|\, s)) Q_k(s, a).$$

  Furthermore, we have

  $$\bar{F}(Q) = M^{\pi_b} H(Q) + (I - M^{\pi_b}) Q,$$

where $M^{\pi_b}$ is a diagonal matrix with diagonal entries $\mu^{\pi_b}(s)\pi_b(a\,|\,s)$, and $H$ is the Bellman optimality operator.

Next, we have

$$[\bar{F}(Q_1) - \bar{F}(Q_2)]_{s,a} = \mu^{\pi_b}(s)\pi_b(a\,|\,s)(H(Q_1) - H(Q_2))_{s,a} + (1 - \mu^{\pi_b}(s)\pi_b(a\,|\,s))(Q_1(s,a) - Q_2(s,a))$$
$$\leq \mu^{\pi_b}(s)\pi_b(a\,|\,s)\|H(Q_1) - H(Q_2)\|_\infty + (1 - \mu^{\pi_b}(s)\pi_b(a\,|\,s))\|Q_1 - Q_2\|_\infty$$
$$\leq \mu^{\pi_b}(s)\pi_b(a\,|\,s)\gamma\|Q_1 - Q_2\|_\infty + (1 - \mu^{\pi_b}(s)\pi_b(a\,|\,s))\|Q_1 - Q_2\|_\infty$$
$$= (1 - (1-\gamma)\mu^{\pi_b}(s)\pi_b(a\,|\,s))\|Q_1 - Q_2\|_\infty$$
$$\leq (1 - (1-\gamma)\rho_b)\,\|Q_1 - Q_2\|_\infty,$$

where in the second inequality we used that the Bellman optimality operator is a $\gamma$-contraction. It follows that

$$\|\bar{F}(Q_1) - \bar{F}(Q_2)\|_\infty \leq (1 - (1-\gamma)\rho_b)\,\|Q_1 - Q_2\|_\infty.$$

Hence, we can write

$$\|\bar{F}(Q_1) - \bar{F}(Q_2)\|_p \leq (|\mathcal{S}||\mathcal{A}|)^{1/p}\|\bar{F}(Q_1) - \bar{F}(Q_2)\|_\infty$$
$$\leq (|\mathcal{S}||\mathcal{A}|)^{1/p}\,(1 - (1-\gamma)\rho_b)\,\|Q_1 - Q_2\|_\infty$$
$$\leq (|\mathcal{S}||\mathcal{A}|)^{1/p}\,(1 - (1-\gamma)\rho_b)\,\|Q_1 - Q_2\|_p.$$

Then, for any

$$p > p_{\min} := \frac{\ln(|\mathcal{S}||\mathcal{A}|)}{\ln\left(\frac{1}{1-(1-\gamma)\rho_b}\right)} \geq 2,$$

Assumption 5.1 is satisfied with $\gamma_c = (|\mathcal{S}||\mathcal{A}|)^{1/p}\,(1 - (1-\gamma)\rho_b)$.

- **Assumption 5.2:** Since the state and action space are both bounded, the noise in the update of $Q$-learning is also bounded. Then, Equation (5.1) is satisfied for some $\sigma > 0$ by Hoeffding's lemma. In addition, Equation (5.2) is satisfied due to Lemma C.2.

- **Assumption 4.1:** This assumption is satisfied with $M = p - 1$ due to Lemma C.4.

- **Assumption 4.2:** Since $\bar{F}$ is a contraction, Banach's fixed point theorem implies it has a unique fixed point $Q^*$, and Lemma C.3 implies that $J_{\bar{F}}(x^*) - I$ is invertible. Moreover, Assumption 6.1 implies that $F$ is linear in a neighborhood of $Q^*$, and thus there exists $R > 0$ such that $F$ is $(0, R)$-locally psuedo smooth with respect to the $\|\cdot\|_p$ norm.

- **Assumption 4.3:** We have

$$|(F(Q^*, w_{k+1}) - \bar{F}(Q^*))_{s,a}| \leq \frac{2R_{\max}}{1-\gamma}\mathbb{1}_{\{s=s_k^0\}}.$$

Hence,

$$|\langle F(Q^*, w_{k+1}) - \bar{F}(Q^*), v\rangle| \leq \|F(V^\pi, w_{k+1}) - \bar{F}(V^\pi)\|_p^*.\|v\|_p \quad \text{(Hölder's inequality)}$$
$$\leq \frac{2R_{\max}}{1-\gamma}\|v\|_p.$$

Then,

$$\mathbb{E}\left[\exp\left(\lambda\langle F(Q^*, w_{k+1}) - \bar{F}(Q^*), v\rangle\right)\big|\mathcal{F}_k\right] \leq \exp\left(\lambda^2\frac{2R_{\max}^2\|v\|_p^2}{(1-\gamma)^2}\right).$$
$$\text{(Hoffding's lemma)}$$

Therefore, the first part of Assumption 4.3 is satisfied with $\bar{\sigma}^2 = 4R_{\max}^2/(1-\gamma)^2$.

On the other hand, since we assume finite state and action spaces, and $w_{k+1}$ can only take finitely many values, therefore, by Hoeffding's lemma there exist $\hat{\sigma} < \infty$ such that the second part of Assumption 4.3 holds.

Furthermore, since $p \geq 2$ and $\|x\|_p \leq \|x\|_2$, we have $u_{c2} = 1$. By a direct application of Theorem 5.1 we have that, with probability at least $1 - \delta$,

$$\|\bar{Q}_k - Q^*\|_p^2 \leq \frac{4R_{\max}^2}{(1-\gamma)^2}\frac{24\log(2/\delta) + 3|\mathcal{S}||\mathcal{A}|}{(1-\gamma_c)^2\,(k+1)} + \mathcal{O}(p)\tilde{\mathcal{O}}\left(\frac{1}{k^{3/2}}\right)\mathcal{O}(\log(1/\delta)).$$

Furthermore, we have

$$\|\bar{Q}_k - Q^*\|_p^2 \geq \|\bar{Q}_k - Q^*\|_\infty^2.$$

Hence, with probability $1 - \delta$, we have

$$\|\bar{Q}_k - Q^*\|_\infty^2 \leq \frac{4R_{\max}^2}{(1-\gamma)^2} \frac{24 \log(2/\delta) + 3|\mathcal{S}||\mathcal{A}|}{(1-\gamma_c)^2 (k+1)} + \mathcal{O}(p)\tilde{\mathcal{O}}\left(\frac{1}{k^{3/2}}\right) \mathcal{O}(\log(1/\delta)). \qquad \text{(B.19)}$$

This result holds for any $p$. In particular, by choosing $p = p_{\min}(k+1)^{1/4}$, we have

$$\frac{1}{1-\gamma_c} = \frac{1}{1 - (|\mathcal{S}||\mathcal{A}|)^{1/p}(1 - (1-\gamma)\rho_b)}$$

$$\leq \frac{1}{(1-\gamma)\rho_b} + \frac{2(1 - (1-\gamma)\rho_b)\ln(|\mathcal{S}||\mathcal{A}|)}{(1-\gamma)^2 \rho_b^2} \frac{1}{p}, \qquad \text{(B.20)}$$

where in the last inequality we used the fact that

$$\frac{1}{1 - ab^x} \leq \frac{1}{1-a} + \frac{2a\ln(b)}{(1-a)^2} x$$

for all $0 \leq x \leq (1-a)/(2a\ln(b))$, $0 < a < 1$ and $b > 1$, and we take $x = 1/p$, $a = (1 - (1-\gamma)\rho_b)$ and $b = |\mathcal{S}||\mathcal{A}|$. Substituting Equation (B.20) in Equation (B.19), we get

$$\|\bar{Q}_k - Q^*\|_\infty^2 \leq \frac{4R_{\max}^2}{(1-\gamma)^2} \frac{24 \log(2/\delta) + 3|\mathcal{S}||\mathcal{A}|}{(k+1)} \left(\frac{1}{(1-\gamma)\rho_b} + \mathcal{O}\left(\frac{1}{k^{1/4}}\right)\right)^2 + \tilde{\mathcal{O}}\left(\frac{1}{k^{5/4}}\right) \mathcal{O}(\log(1/\delta))$$

$$\leq \frac{4R_{\max}^2}{(1-\gamma)^4 \rho_b^2} \frac{24 \log(2/\delta) + 3|\mathcal{S}||\mathcal{A}|}{k+1} + \tilde{\mathcal{O}}\left(\frac{1}{k^{5/4}}\right) \mathcal{O}(\log(1/\delta)).$$

## B.10   Proof of Theorem 6.3

The update of off-policy TD($n$) can be written as follows:

$$v_{k+1} = v_k + \alpha_k\phi(S_k^0) \sum_{l=0}^{n-1} \gamma^l \left[\prod_{j=0}^{l} \frac{\pi(A_k^j|S_k^j)}{\pi_b(A_k^j|S_k^j)}\right] \left[\mathcal{R}(S_k^l, A_k^l) + \gamma\phi(S_k^{l+1})^\top v_k - \phi(S_k^l)^\top v_k\right]$$

$$= v_k + \alpha_k\phi(S_k^0) \sum_{l=0}^{n-1} \gamma^l \left[\prod_{j=0}^{l} \frac{\pi(A_k^j|S_k^j)}{\pi_b(A_k^j|S_k^j)}\right] \left[\gamma\phi(S_k^{l+1})^\top - \phi(S_k^l)^\top\right] v_k$$

$$+ \alpha_k\phi(S_k^0) \sum_{l=0}^{n-1} \gamma^l \left[\prod_{j=0}^{l} \frac{\pi(A_k^j|S_k^j)}{\pi_b(A_k^j|S_k^j)}\right] \mathcal{R}(S_k^l, A_k^l)$$

$$= (1 - \bar{\alpha}_k)v_k + \bar{\alpha}_k(A_k v_k + b_k),$$

where

$$A_k = I + \frac{1}{\xi}\phi(S_k^0) \sum_{l=0}^{n-1} \gamma^l \left[\prod_{j=0}^{l} \frac{\pi(A_k^j|S_k^j)}{\pi_b(A_k^j|S_k^j)}\right] \left[\gamma\phi(S_k^{l+1})^\top - \phi(S_k^l)^\top\right],$$

$$b_k = \frac{1}{\xi}\phi(S_k^0) \sum_{l=0}^{n-1} \gamma^l \left[\prod_{j=0}^{l} \frac{\pi(A_k^j|S_k^j)}{\pi_b(A_k^j|S_k^j)}\right] \mathcal{R}(S_k^l, A_k^l),$$

and $\bar{\alpha}_k = \xi\alpha_k$. Hence, off-policy TD($n$) with linear function approximation can be written in the form of Equation (1.1) with $F(v_k, w_{k+1}) = A_k v_k + b_k$, where $w_{k+1} = [(S_k^i, A_k^i)]_{0 \leq i \leq n}$.

Next, we verify that the assumptions of Theorem 5.1 are satisfied

- **Assumption 5.1:** As shown in [11, Proposition 3.1] we have

$$\bar{F}(v_k) = v_k + \frac{1}{\zeta}\Phi^\top \mathcal{K}^{\pi_b}((\gamma P^\pi)^n \Phi v_k - \Phi v_k),$$

  and for large enough $n$, we have that

$$\Phi^\top \mathcal{K}^{\pi_b}((\gamma P^\pi)^n \Phi - \Phi)$$

  is a Hurwitz matrix. Hence, by [5, Page 46 footnote] there exists $\zeta$ large enough such that $\bar{F}(\cdot)$ is a contraction with respect to some weighted 2-norm. We denote this norm as $\|\cdot\|_c$.
- **Assumption 5.3:**
    (i) As mentioned earlier, $\|\cdot\|_c$ is a weighted 2-norm.
    (ii) Since $F$ is linear, and the state-action space is bounded, this assumption is satisfied for some $L > 0$.
    (iii) This assumption holds due to [52, Lemma 9]
    (iv) Since $F$ is linear, and the state-action space is bounded, this assumption is satisfied for some $\tau > 0$.
    (v) Since $F$ is linear, and the state-action space is bounded, this assumption is satisfied for some $\Sigma \succeq 0$.
- **Assumption 4.1:** Since the norm $\|\cdot\|_c^2$ is a weighted 2-norm, then it is smooth.
- **Assumption 4.2:** Since $\bar{F}$ is a contraction, Banach's fixed point theorem implies it has a unique fixed point, and Lemma C.3 implies that $J_{\bar{F}}(x^*) - I$ is invertible. Moreover, since $F$ is linear, it is $(0, \infty)$-locally psuedo smooth.
- **Assumption 4.3:** Since the noise is bounded, this assumption is satisfied for some $\bar{\sigma}$ and $\hat{\sigma}$.

Applying Theorem 5.1 yields the result.

## C   Technical lemmas

**Lemma C.1.** *Assume $X_1, X_2, \ldots$ is a series of martingale difference random vectors in $\mathbb{R}^d$. In addition, assume $X_i$ to be $\eta_i^2$-subgaussian conditioned on filtration $\mathcal{F}_{i-1}$ generated by the random variables $\{X_1, X_2, \ldots, X_{i-1}\}$. Then $X_1 + X_2 + \cdots + X_n$ is a $\sum_{i=1}^n \eta_i^2$-subgaussian random vector.*

*Proof:* We prove this by induction in the number of terms $n$. The case $n = 1$ is immediate. Now suppose that, for any $d$-dimensional vector $v$, we have

$$\mathbb{E}\left[\exp\left(\lambda\left\langle\sum_{i=1}^{n-1} X_i, v\right\rangle\right)\right] \le \exp\left(\lambda^2\|v\|_c^2 \sum_{i=1}^{n-1}\frac{\eta_i^2}{2}\right). \tag{C.1}$$

Then,

$$\begin{aligned}
\mathbb{E}\left[\exp\left(\lambda\left\langle\sum_{i=1}^{n} X_i, v\right\rangle\right)\right] &= \mathbb{E}\left[\mathbb{E}\left[\exp\left(\lambda\left\langle\sum_{i=1}^{n} X_i, v\right\rangle\right)\bigg|\mathcal{F}_{n-1}\right]\right]\\
&= \mathbb{E}\left[\exp\left(\lambda\left\langle\sum_{i=1}^{n-1} X_i, v\right\rangle\right)\mathbb{E}\left[\exp\left(\lambda\left\langle X_n, v\right\rangle\right)\bigg|\mathcal{F}_{n-1}\right]\right]\\
&\le \mathbb{E}\left[\exp\left(\lambda\left\langle\sum_{i=1}^{n-1} X_i, v\right\rangle\right)\right]\exp\left(\lambda^2\|v\|_c^2\frac{\eta_n^2}{2}\right)\\
&\hspace{5cm}(X_n \text{ is subgaussian})\\
&\le \exp\left(\lambda^2\|v\|_c^2 \sum_{i=1}^{n}\frac{\eta_i^2}{2}\right). \hspace{1cm}(\text{Inductive hypothesis})
\end{aligned}$$

$\square$

**Lemma C.2.** *Suppose we have a random vector $X \in \mathbb{R}^d$ which is $\eta^2$-subgaussian. Then, for all $\lambda < 1/(2\eta^2 u_{c2}^2)$, we have*

$$\mathbb{E}\left[\exp\left(\lambda\|X\|_2^2\right)\right] \leq \frac{1}{(1 - 2\lambda\eta^2 u_{c2}^2)^{d/2}}$$

*Proof:* Since $X$ is subgaussian, we have

$$\mathbb{E}[\exp(\langle X, v\rangle)] \leq \exp\left(\frac{\eta^2\|v\|_c^2}{2}\right) \leq \exp\left(\frac{\eta^2 u_{c2}^2\|v\|_2^2}{2}\right).$$

Hence, for any $\lambda \in (0, 1)$, we have

$$\mathbb{E}\left[\exp\left(\frac{\langle X, v\rangle - u_{c2}^2\|v\|^2\eta^2}{2\lambda}\right)\right] \leq \exp\left(\frac{\eta^2 u_{c2}^2\|v\|^2(\lambda - 1)}{2\lambda}\right). \tag{C.2}$$

We now integrate both sides with respect to the vector $v$. Integrating the left-hand side, we get

$$\int_{-\infty}^{\infty}\int_{-\infty}^{\infty}\cdots\int_{-\infty}^{\infty} \mathbb{E}\left[\exp\left(\frac{\langle X, v\rangle - u_{c2}^2\|v\|_2^2\eta^2}{2\lambda}\right)\right] dv_d dv_{d-1}\ldots dv_1$$

$$\overset{(a)}{=} \mathbb{E}\left[\int_{-\infty}^{\infty}\int_{-\infty}^{\infty}\cdots\int_{-\infty}^{\infty} \exp\left(\frac{\langle X, v\rangle - u_{c2}^2\|v\|_2^2\eta^2}{2\lambda}\right) dv_d dv_{d-1}\ldots dv_1\right]$$

$$= \mathbb{E}\left[\left\{\int_{-\infty}^{\infty}\exp\left(\frac{X_1 v_1 - u_{c2}^2 v_1^2\eta^2}{2\lambda}\right)dv_1\right\}\cdots\left\{\int_{-\infty}^{\infty}\exp\left(\frac{X_d v_d - u_{c2}^2 v_d^2\eta^2}{2\lambda}\right)dv_d\right\}\right]$$

$$= \mathbb{E}\left[\left\{\frac{\sqrt{2\pi\lambda}}{\eta u_{c2}}\exp\left(\frac{\lambda X_1^2}{2\eta^2 u_{c2}^2}\right)\right\}\cdots\left\{\frac{\sqrt{2\pi\lambda}}{\eta u_{c2}}\exp\left(\frac{\lambda X_d^2}{2\eta^2 u_{c2}^2}\right)\right\}\right]$$

$$= \left(\frac{\sqrt{2\pi\lambda}}{\eta u_{c2}}\right)^d \mathbb{E}\left[\exp\left(\frac{\lambda\|X\|_2^2}{2\eta^2 u_{c2}^2}\right)\right]. \tag{C.3}$$

where $(a)$ is due to Fubini's theorem for non negative functions.

Integrating the right-hand side of Equation (C.2), we get

$$\int_{-\infty}^{\infty}\int_{-\infty}^{\infty}\cdots\int_{-\infty}^{\infty} \exp\left(\frac{u_{c2}^2\eta^2\|v\|_2^2(\lambda - 1)}{2\lambda}\right) dv_d dv_{d-1}\ldots dv_1$$

$$= \left\{\int_{-\infty}^{\infty}\exp\left(\frac{u_{c2}^2\eta^2 v_1^2(\lambda - 1)}{2\lambda}\right)dv_1\right\}\cdots\left\{\int_{-\infty}^{\infty}\exp\left(\frac{u_{c2}^2\eta^2 v_d^2(\lambda - 1)}{2\lambda}\right)dv_d\right\}$$

$$= \left(\frac{\sqrt{2\pi\lambda}}{\eta u_{c2}}\right)^d \frac{1}{(1 - \lambda)^{d/2}}. \tag{C.4}$$

Combining equations (C.3), (C.4), and (C.2), we obtain

$$\mathbb{E}\left[\exp\left(\frac{\lambda\|X\|_2^2}{2\eta^2 u_{c2}^2}\right)\right] \leq \frac{1}{(1 - \lambda)^{d/2}}.$$

By a change of variable, we get the result. $\qquad\square$

**Lemma C.3.** *Consider a differentiable operator $\bar{F} : \mathbb{R}^d \to \mathbb{R}^d$ that is a $\gamma_c$-pseudo-contraction with respect to some norm $\|\cdot\|_c$, with fixed point $x^*$. Then, we have*

$$\nu = \min_{z\in\mathbb{R}^d}\left\{\frac{\|(J_{\bar{F}}(x^*) - I)z\|_c}{\|z\|_c}\right\} \geq 1 - \gamma_c.$$

*Proof:* We have

$$\gamma_c \geq \left\| \frac{\bar{F}(x^* + y) - \bar{F}(x^*)}{\|y\|_c} \right\|_c \qquad \text{(Pseudo-contraction property)}$$

$$= \left\| J_{\bar{F}}(x^*) \frac{y}{\|y\|_c} + \tilde{h}(y) \right\|_c \qquad \text{(definition of Jacobian matrix)}$$

$$\geq \left\| J_{\bar{F}}(x^*) \frac{y}{\|y\|_c} \right\|_c - \frac{o(\|y\|_c)}{\|y\|_c}, \qquad \text{(triangle inequality)}$$

where by the definition of the Jacobian, we used the fact that $\|\tilde{h}(y)\|_c = o(\|y\|_c)/\|y\|_c$. By taking the supremum over all vectors $y \in \mathbb{R}^d$, we have

$$\gamma_c \geq \sup_{y \in \mathbb{R}^d} \left\{ \left\| J_{\bar{F}}(x^*) \frac{y}{\|y\|_c} \right\|_c - \frac{o(\|y\|_c)}{\|y\|_c} \right\}.$$

Note that

$$\left\| J_{\bar{F}}(x^*) \frac{y}{\|y\|_c} \right\|_c - \frac{o(\|y\|_c)}{\|y\|_c} \leq \max_{y \in \mathbb{R}^d} \left\{ \left\| J_{\bar{F}}(x^*) \frac{y}{\|y\|_c} \right\|_c \right\},$$

which is attained at some $\tilde{y}$ (because it is equivalent to maximizing a continuous function over a compact set). Moreover, note that for every constant $r \neq 0$, the vector $r\tilde{y}$ also attains the maximum above. Let $y_n = \tilde{y}/n$. Then,

$$\lim_{n \to \infty} \left\| J_{\bar{F}}(x^*) \frac{y_n}{\|y_n\|_c} \right\|_c - \frac{o(\|y_n\|_c)}{\|y_n\|_c} = \max_y \left\{ \left\| J_{\bar{F}}(x^*) \frac{y}{\|y\|_c} \right\|_c \right\} - \lim_{n \to \infty} \frac{o(1/n)}{1/n}$$

$$= \max_{y \in \mathbb{R}^d} \left\{ \left\| J_{\bar{F}}(x^*) \frac{y}{\|y\|_c} \right\|_c \right\}.$$

It follows that

$$\sup_{y \in \mathbb{R}^d} \left\{ \left\| J_{\bar{F}}(x^*) \frac{y}{\|y\|_c} \right\|_c - \frac{o(\|y\|_c)}{\|y\|_c} \right\} = \max_{y \in \mathbb{R}^d} \left\{ \left\| J_{\bar{F}}(x^*) \frac{y}{\|y\|_c} \right\|_c \right\},$$

and thus

$$\gamma_c \geq \max_{y \in \mathbb{R}^d} \left\{ \left\| J_{\bar{F}}(x^*) \frac{y}{\|y\|_c} \right\|_c \right\}. \tag{C.5}$$

Hence,

$$\nu = \min_{z \in \mathbb{R}^d} \left\{ \frac{\|(J_{\bar{F}}(x^*) - I)z\|_c}{\|z\|_c} \right\}$$

$$\geq \min_{z \in \mathbb{R}^d} \left\{ \frac{\|z\|_c - \|J_{\bar{F}}(x^*)z\|_c}{\|z\|_c} \right\} \qquad \text{(triangle inequality)}$$

$$= 1 - \max_{z \in \mathbb{R}^d} \left\{ \frac{\|J_{\bar{F}}(x^*)z\|_c}{\|z\|_c} \right\}$$

$$\geq 1 - \gamma_c. \qquad \text{(Eq. (C.5))}$$

$\square$

**Lemma C.4.** *Let $p \geq 2$, $q = p/(p-1)$, and consider the weights $w = (w_1, \ldots, w_d) \in \mathbb{R}_{>0}^d$. Define the weighted norms*

$$\|x\|_{p,w} := \left( \sum_{i=1}^d w_i |x_i|^p \right)^{1/p}, \qquad \text{and} \qquad \|u\|_{q,w^\#} := \left( \sum_{i=1}^d w_i^\# |u_i|^q \right)^{1/q},$$

*where $w_i^\# := w_i^{-1/(p-1)}$. Let $f(x) = \frac{1}{2}\|x\|_{p,w}^2$. Then, for all $x, y \in \mathbb{R}^d$, we have*

$$\|\nabla f(x) - \nabla f(y)\|_{q,w^\#} \leq (p-1)\|x - y\|_{p,w}.$$

*Proof:* Introduce the diagonal map $T : \mathbb{R}^d \to \mathbb{R}^d$ such that $T(x) := Wx$, where $W$ is a diagonal matrix with that $i$-th diagonal entry equal to $w_i^{1/p}$. Then, we have

$$\|x\|_{p,w} = \|T(x)\|_p, \qquad \text{and} \qquad \|u\|_{q,w\#} = \|T^{-1}(u)\|_q. \tag{C.6}$$

Let $g(z) = \frac{1}{2}\|z\|_p^2$. Since $f = g \circ T$ and $W$ is diagonal, by the chain rule, we have

$$\nabla f(x) = T(\nabla g)(T(x)). \tag{C.7}$$

On the other hand, as shown in [2, Example 5.11], for $p \geq 2$ we have

$$\|\nabla g(z) - \nabla g(z')\|_q \leq (p-1)\|z - z'\|_p, \qquad \forall z, z' \in \mathbb{R}^d. \tag{C.8}$$

Fix $x, y$, and set $z = T(x)$, $z' = T(y)$. We have

$$
\begin{aligned}
\|\nabla f(x) - \nabla f(y)\|_{q,w\#} &= \|T[\nabla g(z) - \nabla g(z')]\|_{q,w\#} && \text{(Eq. (C.7))} \\
&= \|\nabla g(z) - \nabla g(z')\|_q && \text{(Eq. (C.6))} \\
&\leq (p-1)\|z - z'\|_p && \text{(Eq. (C.8))} \\
&= (p-1)\|x - y\|_{p,w}. && \text{(Eq. (C.6))}
\end{aligned}
$$

The chain above yields the desired Lipschitz constant $L = p - 1$:

$$\|\nabla f(x) - \nabla f(y)\|_{q,w\#} \leq (p-1)\|x - y\|_{p,w}.$$

$\square$

