# OpenReview forum: "A General-Purpose Theorem for High-Probability Bounds of Stochastic Approximation with Polyak Averaging"
_NeurIPS.cc/2025/Conference — NeurIPS 2025 poster_

### Official Review · Reviewer_VkGz · 2025-06-26

**Clarity:** 3
**Significance:** 3
**Originality:** 3
**Rating:** 5
**Confidence:** 3

**Summary:**

For stochastic approximation algorithms, given a high probability bound for (unaveraged) iterates, this paper turn it into a high probability bound for the Polyak-Ruppert average. Various implications of the bound are discussed in detail, including tightness, simplifications in different scenarios, and applications to "abstract" contractive operators as well as concrete RL algorithms.

**Questions:**

See ''Strengths And Weaknesses''.

**Ethical Concerns:**

["NO or VERY MINOR ethics concerns only"]

**Final Justification:**

I think this is a solid theoretical paper. My concerns (mainly about the presentation) are well-addressed in the rebuttal. I keep my score 5 and recommend acceptance.

**Limitations:**

See ''Strengths And Weaknesses''.

**Quality:**

3

**Strengths And Weaknesses:**

This is a solid theory paper. Although the main result looks complicated, at the end (Section 5) it brings clear insights on the performance difference between iterates and the average (improving constants in additive noise case and making tails heavier in the multiplicative noise case). The presentation can still be improved (e.g., to let non-SA-expert readers learn more from the paper). Here are some suggestions and questions.

What motivates the generalization of the notion of smoothness in Assumption 4.2 (iii)?

May add an insightful/informal version of Theorem 4.1 that reflects the discussion (between Theorem 4.1 and Corollary 4.1) without giant expressions, to highlight some takeaway about the main result itself, before exploring various implications.

What is $u_{c2}$?

May briefly explain why Assumption 5.2 and 5.3 correspond to additive and multiplicative noise, respectively.

For Section 6, instead of rushing through three settings without discussion, it may be better to focus on one of them, to let readers know what insights these bounds bring in the RL setting.

The last question is a "philosophical" one. This paper treat bounds for $x_t$ as given (first step) and transform them into bounds for $y_t$ (second step), resulting in two completely separate steps. What is the price of this separation? Is it possible to partially merge the two steps to generate better result?

---

> ### Author Rebuttal · Authors · 2025-07-29
>
> We thank the reviewer for their positive evaluation and thoughtful feedback. We are encouraged by your recognition of the technical strength of our results and the insights conveyed in Section 5. We address your specific suggestions and questions below:
>
> (1) Motivation for the generalized notion of smoothness in Assumption 4.2(iii):
>
> Thank you for this important question. The generalization of smoothness in Assumption 4.2(iii) is motivated by the need to analyze the averaged sequence when the operator is not globally smooth. Many practically relevant SA problems (e.g., Q-learning) involve operators that are only locally smooth in a neighborhood of the solution. Our assumption allows for such cases, while still enabling tight control of the error under averaging. We will elaborate on this motivation more clearly in the revised version.
>
> (2) Suggestion for an informal or takeaway version of Theorem 4.1:
>
> We appreciate this suggestion. While Theorem 4.1 is stated in full generality to maximize applicability, we agree it would be valuable to include an informal version or visual summary highlighting key insights. In the revision, we will add a short paragraph immediately following Theorem 4.1 summarizing the takeaways in plain language.
>
>
> (3) What is $u_{c2}$?
>
> As described at the end of page 4, $u_{c2}$ is the constant that relates the 2-norm and the c-norm as follows: $||.||_c\leq u_{c2}||.||_2$.
>
>
> (4) Clarification on Assumption 5.2 and 5.3 (additive vs multiplicative noise):
>
> Thank you for pointing this out. Assumption 5.2 corresponds to additive noise, where the variance of the noise is uniformly bounded regardless of the variable $x$. In contrast, Assumption 5.3 models multiplicative noise, where the noise grows linearly with the magnitude of the iterate. The additive/multiplicative terminology comes from the simple linear SA. In particular, suppose $F(x,w)=Ax+ b(w)$ for a fixed matrix $A$ and a random vector $b(w)$. This corresponds to additive noise since the noise $b(w)$ is added. However, if we have $F(x,w)=A(w)x+ b(w)$, we have the multiplicative noise, as the noisy matrix $A(w)$ is multiplied with the variable x.
> We will make this correspondence explicit in the exposition to aid readers less familiar with SA theory.
>
>
> (5) Suggestion to focus Section 6 more narrowly:
>
> We appreciate this perspective. The current version of Section 6 includes three settings to demonstrate the generality of our framework, but we agree that a deeper discussion of  the results would better illustrate the implications of our bounds in RL. In the revision, we will use the extra page to expand our discussion of the results in this section.
>
> (6) Philosophical question: Can the two-step analysis (base bounds → averaged bounds) be merged?
>
> Thank you for the insightful question. As shown in Proposition 4.2, the leading term of our high probability bounds are tight up to a universal constant. However, one might be able to get better higher order terms by a joint analysis of the two variables $x_k$ and $y_k$. We leave this direction for future research.

---

> > ### Comment · Reviewer_VkGz · 2025-08-05
> > **Thanks**
> >
> > Thanks for the rebuttal. All questions are well addressed. I do not have further concerns.

---

### Official Review · Reviewer_KhvA · 2025-07-02

**Clarity:** 3
**Significance:** 2
**Originality:** 2
**Rating:** 4
**Confidence:** 3

**Summary:**

This paper provides concentration bounds for general stochastic approximation (SA) algorithms under Markov noise, accounting for both linear and nonlinear function approximators. The paper assumes a high-probability bound for the iterate sequence of the stochastic approximation algorithm, and uses this bound to derive high-probability bounds for the Polyak-Ruppert averaged iterate. The derived bounds are then specialized to contractive SA algorithms, including TD and Q-learning. Lower bounds are also derived that nearly match the leading terms in the upper bounds (up to universal constants).

**Questions:**

My main questions and comments are as listed above under "Strengths and Weaknesses". Without a more detailed comparison with the prior art, the significance of the contributions remains unclear to me.

**Ethical Concerns:**

["NO or VERY MINOR ethics concerns only"]

**Final Justification:**

My initial primary concern was in terms of the specific nature of the contribution in relation to a bunch of other papers (that I pointed to) that also establish finite-time rates. The authors have adequately at least explained the key differences in the settings w.r.t. to these papers. As a result, I have raised my score to '4'.

I did not raise my score any further since the authors did not quite explain why the techniques used to establish high-probability bounds for base iterates cannot be adapted to *averaged* iterates, the latter being the focus of their paper.

**Limitations:**

Yes.

**Paper Formatting Concerns:**

No concerns.

**Quality:**

3

**Strengths And Weaknesses:**

**Strengths**

- Although there has been a lot of recent interest in establishing finite-time rates for SA algorithms, most rates are derived in the mean-squared sense. In contrast, this work establishes high-probability bounds for a fairly general class of SA algorithms, accounting for both linear and nonlinear function approximation, and both additive and multiplicative noise.

- While most papers only provide upper bounds, this work also establishes nearly matching lower bounds.

- The paper is very well structured, and provides a detailed treatment of various special cases (e.g., TD and Q learning) that follow as corollaries of the main result.

**Weaknesses and Comments**

- To derive their general result on the averaged iterate sequence $y_k$, this paper assumes a high-probability bound on the basic iterate sequence $x_k$ to begin with. However, given the generality of the setting considered in this paper, it was quite unclear to me how challenging it would be to obtain high-probability bounds on the $x_k$ sequence to begin with. For instance, say one considers the special case of TD learning with linear function approximation under Markov noise. Obtaining high-probability bounds for the basic iterate sequence $x_k$ seems quite non-trivial in the first place, especially without the assumption of a projection step.

Thus, my main concern is whether the difficulty of obtaining tight concentration bounds for $x_k$ might dominate that for $y_k$.

- Some recent papers have, in fact, established finite-time high probability bounds for TD learning. See References [R1] and [R2] below that are not currently discussed in the paper. Each of these papers establish concentration bounds under Markov noise. As such, I would like a clear discussion of how the results in this paper compare with those in [R1] and [R2].

[R1] Adapting to mixing time in stochastic optimization with Markovian data, Dorfman and Levy, ICML 2022.

[R2]  Finite time analysis of temporal difference learning with linear function approximation: Tail averaging and regularisation, Patil et al., AISTATS 2023.

- Related to the above point, the paper does not do a good job of precisely pin-pointing how their results improve upon/generalize those in the long list of papers reviewed in the Related Works section. In particular, I would like a more transparent comparison with references [18], [32], and [R3] below, all of which do establish high-probability bounds for linear SA.

[R3] High-probability sample complexities for policy evaluation with linear function approximation, Li et al., TIT, 2024.

As far as I can tell, the main (and perhaps the only) difference with the above papers seems to be that the current work allows for nonlinear function approximators. However, the extension from linear to nonlinear SA schemes has also been explored in a variety of works, under suitable regularity conditions on the nonlinearity. Thus, the paper would benefit from a discussion of the new technical challenges that arise in their analysis, relative to what has not appeared before in prior work.

- The results in Theorems 6.1. and 6.2. seem to be for tabular TD and Q-learning, respectively. Aren't these results already subsumed by those in References [28], [29], and [37]? In particular, References [28] and [29] have already established tight upper and lower bounds for tabular Q-learning, while accounting for asynchronous Markov sampling. As such, how does Theorem 6.2. in this paper compare with the results in these references?

- In addition to References [R1]-[R3] above (that are highly relevant for this paper), some other relevant references on TD learning/linear SA that are missing are as follows:

[R4] Temporal Difference Learning as Gradient Splitting, Liu and Olshevsky, ICML 2021.

[R5] Is Temporal Difference Learning Optimal? An Instance-Dependent Analysis, Khamaru et al., SIMODS 2021.

[R6] Accelerated and instance-optimal policy evaluation with linear function approximation, Li et al., SIMODS 2023.

[R7] A Simple Finite Time Analysis of TD Learning with Linear Function Approximation, Mitra, TAC 2024.

The authors should add and discuss the missing references [R1] - [R7] in their revised version.

---

> ### Author Rebuttal · Authors · 2025-07-28
>
> We thank the reviewer for the careful reading and thoughtful comments. We are encouraged by the recognition that our results provide general high-probability bounds for stochastic approximation (SA) algorithms under broad assumptions---particularly accounting for nonlinearity and multiplicative noise---and for the appreciation of our structural clarity and treatment of special cases. Below, we address the main concerns:
>
> (1) On the assumption of high-probability bounds for the base SA iterates.
>
> We agree that deriving high-probability bounds for general SA iterates, especially under Markovian noise and without projection, is a challenging task. However, the focus of our work is not to re-establish such bounds, but to provide a \emph{general framework} that transfers existing high-probability bounds on raw iterates into high-probability guarantees for the \emph{averaged} sequence.
>
> Our framework is \emph{modular}: any bound of the form $\mathbb{P}(\|x_i - x^*\|^2_c \leq \alpha_i f(\delta, k)) \geq 1 - \delta \quad \text{for all } 0 \leq i \leq k$
> can be used as input to our main result (e.g., as derived in [15, 27]). Section 5 demonstrates this instantiation for both additive and multiplicative noise settings.
>
> In the context of reinforcement learning (RL), many works have established high-probability bounds for TD and Q-learning algorithms. Our Section 6 builds directly upon these results to derive \emph{new distribution-level bounds for the averaged iterates}.
>
> It is worth emphasizing that \emph{analyzing the average of SA iterates is fundamentally more delicate} than analyzing the raw iterates. While the base SA sequence with step size $\alpha_k = \alpha / (k + h)^\xi$ achieves an $\mathcal{O}(1/k^\xi)$ rate, the averaged sequence resembles an SA with step size $1/k$, and achieving a tight $\mathcal{O}(1/k)$ concentration requires new analysis techniques.
>
> Finally, we note that our current work focuses on the \emph{martingale noise setting}. While we believe our ideas extend naturally to the \emph{Markovian noise setting}, that is beyond the current scope and will be addressed in a future work. Importantly, our result acts as a \emph{black-box averaging layer} that can be applied to any future high-probability bound on the base iterates, which is where the generality of our approach lies. We will clarify this point in Section 4 of the revised version.
>
> (2) Comparison with [18, 28, 29, 32, 37] and [R1]--[R7].
>
> We thank the reviewer for highlighting these important references. Below, we outline how our work differs from each:
>
>     (a) [28, R4, R1, R6, R7]: These works provide bounds on the \emph{mean squared error}, rather than \emph{high-probability guarantees}.
>
>     (b) [R2]: Studies TD-learning with a \emph{projection step}. In contrast, our results apply to TD-learning without any projection.
>
>     (c) [18, 29, 32, R3, R5]: These works provide \emph{$\delta$-dependent high-probability bounds}, where the step size is explicitly tuned to the fixed $\delta$ level. As a result, they do \emph{not provide a bound on the entire tail} of the distribution. In contrast, our framework supports \emph{$\delta$-independent step sizes}, yielding a bound that holds for \emph{all} $\delta \in (0, 1)$---a much stronger and more flexible guarantee.
>
>     (d) [37]: Analyzes Q-learning without averaging. To compensate for the lack of averaging, it requires step sizes that depend on \emph{unknown problem parameters} such as $\mu_{\min}$. Our result avoids this by focusing on \emph{Polyak–Ruppert averaging}, which permits robust choice of step sizes and yields better high-probability bounds.
>
> In short, the goal and scope of our paper---namely, providing \emph{tight, high-probability, distribution-level bounds for the averaged SA iterates}---are fundamentally different from those of the prior works listed above. Our analysis requires a \emph{new technical machinery} to handle the subtle interplay between averaging and concentration, especially in the presence of nonlinear dynamics and multiplicative noise.
>
> In the next version of the paper we will incorporate [R1]--[R7] into the Related Work section and include a detailed comparison table to more explicitly position our results within the literature.

---

> > ### Comment · Reviewer_KhvA · 2025-08-01
> > **Thanks for the rebuttal, raising my score.**
> >
> > Dear Authors,
> >
> > Thank you for your rebuttal and for addressing my concerns. I will raise my score to '4'.
> >
> > For the revised version of your paper, I would recommend that you incorporate the following:
> >
> > (1) Explain more precisely what challenges appear in establishing concentration bounds for the averaged sequences that cannot be addressed using the known machinery for non-averaged base iterates.
> >
> > (2) On a similar note, highlight more explicitly the fact that your results provide distributional guarantees on the averaged iterate, as opposed to guarantees that hold for a fixed confidence level.

---

> > > ### Author Response · Authors · 2025-08-06
> > >
> > > Thank you for your helpful suggestions and for reconsidering your score. In the revised version of the paper, we will use the additional page to incorporate the points you raised. Specifically, we will (1) explain in more detail the technical challenges in deriving concentration bounds for averaged iterates and why existing tools for non-averaged iterates are insufficient, and (2) more explicitly emphasize that our results provide distributional guarantees rather than fixed-confidence bounds.

---

### Official Review · Reviewer_RYGS · 2025-07-03

**Clarity:** 4
**Significance:** 4
**Originality:** 4
**Rating:** 6
**Confidence:** 3

**Summary:**

This paper considers the problem of finding fixed-points with
stochastic approximation. This general mathematical framework
encompasses a multitude of practical AI settings, including
reinforcement learning. I should note that stochastic approximation is
outside my core research area.

Polyak-Ruppert (PR) averaging was major development in stochastic
approximation, improving on the Robbins-Monro algorithm. It has been
known for a long time that Polyak averaging can achieve asymptotic
optimal convergence rates, usually on the form O(log(1/delta)/k) where
1-delta is the probability of exceeding the error bound and k is the
number of iterations of the algorithm. Recent research has shown
non-asymptotic, finite-time results for the expectation. However, in
practical applications, rather than knowing the expected error, it is
more useful to know the probability that a run of the algorithm will
deviate far from the optimum, i.e., to prove tail-inequalities on the
error distribution of Polyak-Ruppert averaging.

**Questions:**

The proofs are rather involved. I have checked most proofs in detail
without spotting any errors. I have a few suggested changes below.

Minor comments:

In a few places, the paper multiplies big-Oh expressions, e.g., in
line 163. I am not sure if this is conventional in this community,
however, in other places, one would write O(log(1/delta)/k) rather
than O(1/k)O(log(1/delta)).

There are frequent typographic issues with multiplication, where the
cdot is on the baseline, e.g., in the latex environment below
line 220.

Line 778: Replace "easy to see that" with an explanation.

Line 790: Would it be easier to set the \varrho variable directly to 1
here, rather than later?

Line 810: I do not see why O(lambda) \le O(lambda^2) is a
contradiction, even if lambda \lt 1.

Line 826: Add some explanations to these derivations.

Line 842: By Markov's inequality

Line 1004: Explain why the assumptions of Fubini's theorem are
satisfied.

**Ethical Concerns:**

["NO or VERY MINOR ethics concerns only"]

**Final Justification:**

I am satisfied with the rebuttal and remain confident that this is strong result that deserves to be published.

**Limitations:**

Yes

**Quality:**

4

**Strengths And Weaknesses:**

Based on my limited knowledge of stochastic approximation, this paper
makes a breakthrough in this direction. The main result (Theorem 4.1)
is a fixed-time, non-asymptotic upper bound on the error made by the
algorithm. With an accompanying lower bound for a special case, the
authors show that the bound is tight up to a constant factor. The
assumptions for the upper bound are quite relaxed. Several example
applications are made, including the first bound on the error
distributions of averaged TD-learning, averaged Q-learning, and
others.

In overall, the paper is very well presented, including a sufficient
discussion of related work. I have only spotted a few grammatical
issues.

---

> ### Author Rebuttal · Authors · 2025-07-29
>
> We thank the reviewer for their careful reading, detailed engagement with the technical content, and generous assessment of the paper's significance, clarity, and originality. We are encouraged by your recognition of Theorem 4.1 as a breakthrough result in establishing high-probability, non-asymptotic guarantees for Polyak–Ruppert averaging, and for noting the value of our contributions to reinforcement learning applications such as TD and Q-learning.
> We respond below to your comments and suggestions:
>
> (1) Use of multiplicative $\mathcal{O}(\cdot)$ notation (e.g., line 163):
> Thank you for pointing this out. We intentionally wrote the upper bound in this format to split the dependency on $\delta$ and $k$ and make them more clear. Mathematically there is no mistake in our notation. If the reviewer thinks it is confusing, we can change it to just one $\mathcal{O}(\cdot)$ term.
>
> (2) Typographic issues with multiplication:
> We appreciate your careful reading. We will fix them in the next version of the paper.
>
> (3) Line 778 (“easy to see that”):
> We will revise this line to include a brief justification instead of relying on the phrase “easy to see,” which can be opaque to readers. We agree that this will improve clarity and accessibility.
>
> (4) Line 790: Choice of $\varrho$:
> We thank the reviewer for this suggestion. We will update the exposition accordingly to streamline the argument.
>
> (5) Line 810: Clarification on contradiction:
> Having $\mathcal{O}(\lambda) \leq \mathcal{O}(\lambda^2)$ for all $\lambda>0$ is indeed a contradiction. In particular, as $\lambda\rightarrow 0, \mathcal{O}(\lambda^2)$ converges to zero faster than $\mathcal{O}(\lambda)$.
>
> (6) Line 826: Add explanation to derivation:
> We will expand this step with intermediate calculations and justification to help the reader follow the derivation more easily.
>
> (7) Line 842 (“By Markov’s inequality”):
> Thank you for catching this. We will fix it in the next version.
>
> (8) Line 1004: Use of Fubini’s theorem:
> We will add a justification for the application of Fubini’s theorem here, including verification of the integrability condition that allows the interchange of expectation and integral.
>
> We greatly appreciate your feedback and suggestions, which will be incorporated into the revised version to further enhance clarity and rigor.

---

### Official Review · Reviewer_Mrqo · 2025-07-07

**Clarity:** 2
**Significance:** 3
**Originality:** 3
**Rating:** 4
**Confidence:** 3

**Summary:**

The paper presents a general framework to derive finite-time and high-probability performance bounds for Polyak-Ruppert Averaging (an averaged sequence of iterates in a stochastic approximation algorithm). While the paper makes some key assumptions, the utility of SA-based algorithms is immense; therefore, the results are potentially useful in a wide variety of ML and optimization applications. The crucial point is that the paper does not present new algorithms (or motivate new applications) but provides a tight and general framework for analysis that has wide utility.

**Questions:**

please see above

**Ethical Concerns:**

["NO or VERY MINOR ethics concerns only"]

**Final Justification:**

i raised by score to 4.
The only concern with the paper is that it requires quite a bit of rewriting, as described in the author's rebuttal, to become more readable and accessible. I do not have other strong concerns if this paper is going to be accepted as it is overall a good contribution.

**Limitations:**

The paper is hugely limited by the presentation and planning of the technical content. Perhaps, the good goal would be to make the primary theorems readable.

**Quality:**

3

**Strengths And Weaknesses:**

S:

1)The results in the paper are significant, especially since they establish a general performance guarantee of the SA-based algorithm and explain its successes.

W:

1) The paper is technically dense and largely theoretical. My personal opinion is that NeurIPS may not be the best venue for this work (given the reviewing timelines and reviewer load).

2) My primary concern is how Theorem 4.1 (the main technical result ) is presented. While Lines 139 to 147 are too dense and hard to parse, Lines 28 to 35 are too vague and leave more doubts. A strong suggestion is that (in order to make the results more readable and hence usable) the authors may consider replacing the constants in lines 142-144 with symbols and explaining the symbols later. I was hoping to understand how the main bound degrades with f(\delta,k) (iterate-wise concentration).

3)In most SA algorithms f(\delta,k) would be fairly large for small k. As a result, it is not clear what the guarantee would mean as a function of k.

4) It is also surprising that the right-hand side of the result in Theorem 4.1 does not depend on f(\delta, k'), where k'< k (i.e., point-wise errors in the older iterates). Can the authors comment on this?

5)In my opinion, the most usable applications are in section 5, but they are based on additional assumptions. On its own, Assumption 5.3 is confusing, and it is not clear what it means.  e.g. what is "c"?

6)Can the authors explain what "the first" is in Theorem 6.1? I thought the finite time error bounds for TD learning had been established already. It is likely I am missing something. What does "the entire distribution mean" A similar claim is made for Theorem 6.2

---

> ### Author Rebuttal · Authors · 2025-07-28
>
> We thank the reviewer for their thoughtful evaluation and recognition of the technical significance and generality of our results. We address the concerns and suggestions below in order:
>
> (1) [Clarity of Theorem 4.1: Dense presentation vs vague summary]
> We appreciate the suggestion to improve the clarity of Theorem 4.1. In the camera-ready version, we will revise the presentation of the bound by replacing multi-line constants (lines 142–144) with concise symbolic expressions and deferring detailed definitions and decompositions to a subsection immediately following the theorem. We agree that this will make the result easier to digest, especially for readers less familiar with high-probability bounds.
> Regarding the dependency of the upper bound on f (\delta, k) function, we have a detailed discussion in section 4.3 of the paper. We will also enhance this section with additional clarifications to further aid the reader.
>
> (2) [Interpretation of dependence on f(δ, k)]
> As noted in line 159, in most SA algorithms the function $f(\delta, k)$ typically grows at most polylogarithmically in $k$. Consequently, its impact on the higher-order terms in our bound affects the $k$-dependence only logarithmically.
> In Theorem 4.1, we assume $||x_i-x^*||^2\leq \alpha_i f(\delta’,k) \forall i\leq k$. Therefore, while the function $f(\delta', k)$ is uniform across the iterates, the dependence on $i \leq k$ appears through the decreasing step sizes $\alpha_i$. We will include a more detailed explanation of this point in the revised version.
>
>
> (3) [Usability of applications; clarification of Assumption 5.3]
> Thank you for raising this point. Assumption 5.3 is a standard condition in the literature on SA with multiplicative noise (e.g., see [10], [12]). The constant “c” refers to the norm $||\cdot||_c$ used throughout the paper. We will revise the phrasing of Assumption 5.3 to make this clearer and add an illustrative example, such as off-policy TD learning with linear function approximation, to highlight when this assumption applies.
>
> (4) [Clarification on “the first” in Theorem 6.1 and 6.2]
> We appreciate the opportunity to clarify. There do exist prior finite-time bounds for TD and Q-learning in various settings. Our claim in Theorems 6.1 and 6.2 is not about being the first to provide finite-time bounds in general, but rather about establishing the first non-asymptotic high-probability bounds on the entire distribution (i.e., for all $\delta \in (0,1)$) of the error of the averaged iterates.
> (a)	Prior works such as [18], [42] provide bounds for the raw iterates or use step sizes that depend on a fixed confidence level $\delta$, and thus only give pointwise control (i.e., one quantile of the error distribution).
> (b)	In contrast, our bounds are uniform in $\delta$, as our step size is independent of $\delta$, and we obtain bounds that control the entire tail of the error distribution for the averaged iterates. We will revise the phrasing in Theorems 6.1 and 6.2 to clarify this distinction.
>
> (5) [Overall presentation and accessibility]
> We thank the reviewer for acknowledging the technical depth and generality of our results. We are actively working on restructuring Section 4 and its subsections for improved readability.

---

> > ### Comment · Reviewer_Mrqo · 2025-08-01
> >
> > Thanks for your rebuttal and clarifications. The changes being proposed will definitely make this paper stronger.

---

### Note · Authors · 2025-08-13

We sincerely thank all reviewers for their careful evaluations, thoughtful comments, and constructive suggestions. We are encouraged by the positive feedback and are grateful for the opportunity to clarify key aspects of our contribution. In the revised version, we will incorporate all suggested improvements, including clarifying the presentation of Theorem 4.1, enhancing the discussion of technical assumptions, improving exposition in Section 6, and expanding comparisons with prior work in the related literature. We are especially appreciative of the recognition of the generality and novelty of our high-probability bounds for averaged stochastic approximation, as well as their implications for reinforcement learning. Thank you again for your engagement and feedback throughout the review process.

---

### Decision · Program_Chairs · 2025-09-17

**Decision:**

Accept (poster)

**Comment:**

The paper provides a general-purpose theoretical framework for turning high-probability bounds about convergence of individual iterates in stochastic approximation algorithms to high-probability bounds for convergence of average iterates. The paper presents both upper and lower bounds, which are nearly matching. There was uniform support for accepting the paper, with comments and suggestions primarily directed at improving clarity of presentation, which the authors have committed to addressing in a revision.